# Targeted inhibition of STAT/TET1 axis as a therapeutic strategy for acute myeloid leukemia

Xi Jiang[1,2,3], Chao Hu[1,2,4], Kyle Ferchen[1], Ji Nie[5], Xiaolong Cui[5], Chih-Hong Chen[6], Liting Cheng[7], Zhixiang Zuo[1,8], William Seibel[9], Chunjiang He[10], Yixuan Tang[7], Jennifer R. Skibbe[1], Mark Wunderlich[11], William C. Reinhold[12], Lei Dong[1,3], Chao Shen[1,3], Stephen Arnovitz[2], Bryan Ulrich[2], Jiuwei Lu[1], Hengyou Weng[1,2,3], Rui Su[1,3], Huilin Huang[1,3], Yungui Wang[1,2,4], Chenying Li[1,3,4], Xi Qin[1,3], James C. Mulloy[11], Yi Zheng[11], Jiajie Diao [1], Jie Jin[4], Chong Li[7], Paul P. Liu[13], Chuan He[5], Yuan Chen[6] & Jianjun Chen[1,2,3]

Effective therapy of acute myeloid leukemia (AML) remains an unmet need. DNA methyl-cytosine dioxygenase Ten-eleven translocation 1 (TET1) is a critical oncoprotein in AML. Through a series of data analysis and drug screening, we identified two compounds (i.e., NSC-311068 and NSC-370284) that selectively suppress *TET1* transcription and 5-hydroxymethylcytosine (5hmC) modification, and effectively inhibit cell viability in AML with high expression of *TET1* (i.e., *TET1*-high AML), including AML carrying t(11q23)/MLL-rearrangements and t(8;21) AML. NSC-311068 and especially NSC-370284 significantly repressed *TET1*-high AML progression in vivo. UC-514321, a structural analog of NSC-370284, exhibited a more potent therapeutic effect and prolonged the median survival of *TET1*-high AML mice over three fold. NSC-370284 and UC-514321 both directly target STAT3/5, transcriptional activators of *TET1*, and thus repress *TET1* expression. They also exhibit strong synergistic effects with standard chemotherapy. Our results highlight the therapeutic potential of targeting the STAT/TET1 axis by selective inhibitors in AML treatment.

[1] Department of Cancer Biology, University of Cincinnati, Cincinnati, OH 45219, USA. [2] Section of Hematology/Oncology, Department of Medicine, University of Chicago, Chicago, IL 60637, USA. [3] Department of Systems Biology, Beckman Research Institute of City of Hope, Monrovia, CA 91016, USA. [4] Department of Hematology, The First Affiliated Hospital, Zhejiang University, Hangzhou, Zhejiang 310003, China. [5] Department of Chemistry, Department of Biochemistry and Molecular Biology, Institute for Biophysical Dynamics, Howard Hughes Medical Institute, University of Chicago, Chicago, IL 60637, USA. [6] Department of Molecular Medicine, Beckman Research Institute of City of Hope, Duarte, CA 91010, USA. [7] Key Laboratory of Luminescence and Real-time Analytical Chemistry (Ministry of Education), College of Pharmaceutical Sciences, Southwest University, Chongqing 400715, China. [8] Sun Yat-sen University Cancer Center, State Key Laboratory of Oncology in South China, Collaborative Innovation Center for Cancer Medicine, Guangzhou 510060, China. [9] Division of Oncology, Cincinnati Children's Hospital Medical Center, Cincinnati, OH 45229, USA. [10] School of Basic Medical Sciences, Wuhan University, Wuhan 430071, China. [11] Experimental Hematology and Cancer Biology, Cincinnati Children's Hospital Medical Center, Cincinnati, OH 45229, USA. [12] Developmental Therapeutics Branch, Center for Cancer Research, NCI, NIH, Bethesda, MD 20892, USA. [13] Genetics and Molecular Biology Branch, National Human Genome Research Institute, NIH, Bethesda, MD 20892, USA. Correspondence and requests for materials should be addressed to X.J. (email: xjiang@coh.org) or to J.C. (email: jianchen@coh.org)

Acute myeloid leukemia (AML) is one of the most common and fatal forms of hematopoietic malignancies[1–4]. Despite the improved risk stratifications and treatment-adapted strategies, with standard chemotherapies, still only 35–40% of younger (aged<60) and 5–15% of older (aged≥60) patients with AML can survive over 5 years[4,5]. Many AML subtypes, such as the *MLL*-rearranged AMLs, are often associated with unfavorable outcome[1,6,7]. Further, current treatment frequently involves intensive post-remission treatment with multiple cycles of high-dose cytarabine (Ara-C), which impairs the quality of life of the patients[8]. While the incidence of AML is continually rising due to aging[9], most elderly patients cannot bear intensive chemotherapy and are associated with very poor survival[4,5]. Thus, improved therapeutic strategies with less intensive treatment but a higher cure rate are urgently needed.

The Ten-eleven translocation (TET) proteins (including TET1/2/3) are known to be able to convert 5-methylcytosine (5mC) to 5-hydroxymethylcytosine (5hmC), leading to DNA demethylation[10,11]. *TET1*, the founding member of the *TET* family, was first identified as a fusion partner of the *MLL* gene associated with t (10;11)(q22;q23) in AML[12,13]. In contrast to the repression and tumor-suppressor role of TET2 observed in hematopoietic malignancies[14–17], we recently showed that *TET1* was significantly upregulated in *MLL*-rearranged AML and played an essential oncogenic role in the development of MLL-fusion-induced leukemia[18,19]. An independent study by Zhao et al. confirmed the essential oncogenic role of Tet1 in the development of myeloid malignancies[20]. Thus, given the fact that knockout of *Tet1* expression shows only very minor effects on normal development including hematopoiesis[21], TET1 is an attractive therapeutic target for AML.

In the present study, through a series of in vitro drug screening and in vivo preclinical animal model studies, we identified chemical compounds NSC-370284 and UC-514321 (a more effective analog of NSC-370284) as potent inhibitors that significantly and selectively suppress the viability of AML cells with high level of *TET1* expression (i.e., *TET1*-high AML cells), and dramatically repress the progression of *TET1*-high AML in mice. These compounds directly bind STAT3/5 as STAT inhibitors and thereby suppress *TET1* transcription and TET1 signaling, leading to potent anti-leukemic effects.

## Results

### NSC-311068 and 370284 inhibit *TET1*-high AML cell viability.
We previously reported the high expression and oncogenic role of *TET1* in AML[18,19]. In fact, high expression of *TET1* was found not only in AML, but also in various tumors including uterine cancer, glioma, etc., and especially, in testicular germ cell malignancies (Supplementary Fig. 1). This indicates potential oncogenic role of *TET1* in many cancers where *TET1* expression level is relatively high. In order to identify chemical compounds that may target TET1 signaling, we searched the drug-sensitivity/gene expression database of a total of 20,602 chemical compounds in the NCI-60 collection of cancer cell samples[22]. We found the expression levels of endogenous *TET1* showed a significant positive correlation with the responsiveness of cancer cells across the NCI-60 panel to 953 compounds ($r > 0.2$; $P < 0.05$). We selected the top 120 with the highest $r$ values and tested their effects on cell viability of a *TET1*-high AML cell line, i.e., MONOMAC-6/t(9;11). Then, the top 20 showing the most significant inhibitory effects (Supplementary Tables 1–2) were further tested in three other *TET1*-high AML cell lines including THP-1/t(9;11), KOCL-48/t(4;11), and KASUMI-1/t(8;21) AML cells, along with MONOMAC-6 cells as a positive control (Supplementary Fig. 2a–e and Supplementary Table 3). Actually, we

found that *TET1* is highly expressed not only in *MLL*-rearranged AML as we reported previously[19], but also in AML carrying t (8;21); moreover, depletion of *Tet1* expression also significantly inhibited t(8;21) fusion gene-induced colony-forming/replating capacity of mouse bone marrow (BM) progenitor cells (Supplementary Fig. 3). Our results showed that NSC-311068 (6-(1-Pyrrolidinyl(3,4,5-trimethoxyphenyl)methyl)-1,3-benzodioxol-5-ol; $C_{21}H_{25}NO_6$) and NSC-370284 (Pyrimidine, 4-[(2,4-dinitrophenyl)thio]-; $C_{10}H_6N_4O_4S$) exhibited the most significant effects in inhibiting cell viability of all four *TET1*-high AML cell lines, whereas showing no significant inhibition on viability of NB4/t(15;17) AML cells, a control cell line with very low level of *TET1* expression (Fig. 1a, b). In the NCI-60 collection, cell lines with relatively higher *TET1* expression levels showed more obvious positive correlation between *TET1* expression level and activity of both NSC-311068 and NSC-370284, compared to that across the entire NCI-60 panel, whereas cell lines with relatively lower *TET1* expression levels exhibited no obvious positive correlation (NSC-311068) or even negative correlation (NSC-370284) (Supplementary Table 2c, d). In *TET1*-high AML cells, NSC-311068 and 370284 significantly repressed the level of *TET1* expression (Fig. 1c), as well as the global 5hmC level (Fig. 1d). In order to rule out the possibility of non-specific toxicity, we reduced the dose of NSC-311068 and NSC-370284 to 25 nM, and tested gene expression and cell viability 24 h after treatment. The low dose, short-term treatments again resulted in a significant downregulation of *TET1* transcription, accompanied with a very minor decrease in the viability of MONOMAC-6, THP-1, and KOCL-48 cells (Fig. 1e, f). Thus, it is unlikely that the inhibitory effects of NSC-311068 and NSC-370284 on *TET1* expression were due to nonspecific toxicity.

### NSC-311068 and 370284 suppress AML progression in vivo.
The potential in vivo therapeutic effects of NSC-311068 and 370284 were then tested with the *MLL-AF9* AML model. NSC-311068 and especially 370284 treatments significantly inhibited *MLL-AF9*-induced AML in secondary bone marrow transplantation (BMT) recipient mice, by prolonging the median survival from 49 days (control) to 94 (NSC-311068) or >200 (NSC-370284) days (Fig. 2a). Notably, 57% (4 out of 7) of the NSC-370284-treated mice were cured, as the pathological morphologies in peripheral blood (PB), BM, spleen, and liver tissues all turned to normal (Fig. 2b). The in vivo downregulation of Tet1 expression by the compounds at both RNA and protein levels was validated by qPCR (Fig. 2c) and western blotting (Fig. 2d; Supplementary Fig. 13a, b), respectively. In another AML model induced by *AML-ETO9a* (AE9a)[23], NSC-311068 and NSC-370284 also exhibited remarkable therapeutic effects, with an elongated median survival from 46 days (control) to 95 (NSC-311068) and 122 (NSC-370284) days, respectively (Fig. 2e). Interestingly, NSC-370284 has been reported previously as an analog of the natural product podophyllotoxin (PPT) that was associated with anti-leukemic activity[24,25]. Given the better therapeutic effect of NSC-370284 in vivo (Fig. 2a, b, e), we decided to focus on NSC-370284 for further studies.

### NSC-370284 targets STAT3/5 and suppresses *TET1* expression.
To decipher the molecular mechanism by which NSC-370284 represses *TET1* expression, we adapted the strategy developed by Kapoor and colleagues[26] to identify direct target protein(s) of NSC-370284. Briefly, multiple-drug-resistant clones were established and transcriptome sequencing was conducted to find mutations in each clone; the assumption was that the critical components of the signaling of the drug target(s) would have a high chance to carry mutations in drug-resistant clones[26]. To this

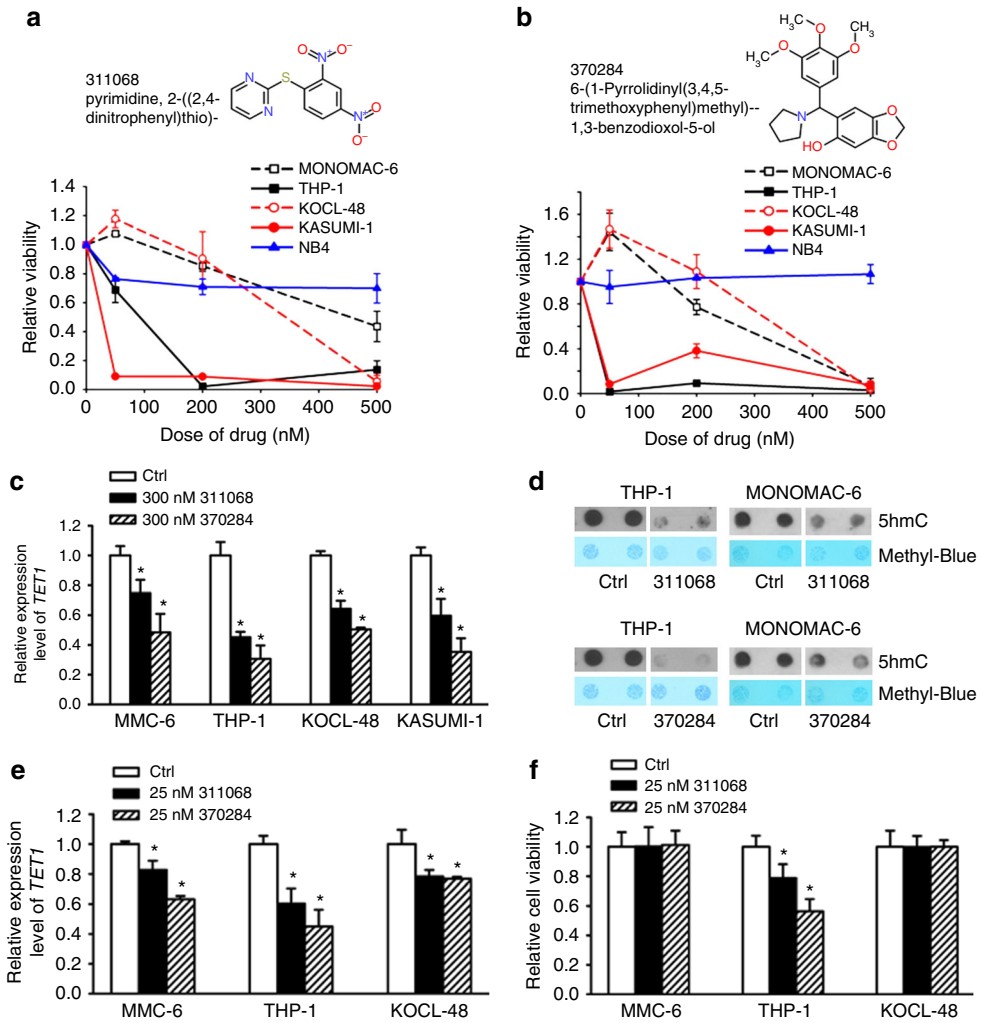

**Fig. 1** NSC-311068 and NSC-370284 suppress the viability of AML cells with high *TET1* level. **a**, **b** *TET1*-high AML cell lines including MONOMAC-6, THP-1, KOCL-48, and KASUMI-1, along with a *TET1*-low control cell line (i.e., NB4), were treated with NSC-311068 (**a**), or NSC-370284 (**b**), at indicated doses (0, 50, 200, 500 nM). Cell viability was analyzed by MTS 48 h post treatment. **c** Repression of *TET1* expression by NSC-311068 and NSC-370284 in AML cell lines. Cells were treated with DMSO, or 300 nM NSC-311068 or NSC-370284. *TET1* expression levels were detected by qPCR 48 h post treatment. **d** NSC-311068 and NSC-370284 (both at 300 nM) repressed global 5hmC level in THP-1 (left panels) and MONOMAC-6 (right panels) cells. **e**, **f** MONOMAC-6, THP-1, and KOCL-48 cells were treated with DMSO, or 25 nM NSC-311068 or NSC-370284. *TET1* expression levels (**e**), and cell viability (**f**), were detected 24 h post treatment. *$P < 0.05$, two-tailed *t*-test. MMC-6, MONOMAC-6. Error bar indicates SD of triplicate experiments

end, we treated THP-1 AML cells with high to moderate concentration of NSC-370284 for over 100 days and then isolated a set of individual drug-resistant THP-1 single clones (see the representitives in Supplementary Fig. 4a). These drug-resistant cells showed no significant downregulation of *TET1* expression upon treatment of NSC-370284 (Supplementary Fig. 4b). Through RNA-seq of 6 of the NSC-370284-resistant clones, recurrent mutations were found in 14 genes in at least two individual clones. Ingenuity pathway analysis (IPA) was used to analyze biological relationships amongst the 14 mutated genes (Supplementary Table 4). The top five networks identified by IPA, based on Fisher's exact test, are associated with cancer, hematological disease, immunological disease, etc. (Supplementary Table 5). The top one network identified by IPA involving all of the 14 genes is closely associated with the JAK/STAT5 pathway (Fig. 3a). A number of these genes have been reported to be associated with the JAK/STAT signaling[27–35]. It is likely that mutations in such genes may overcome NSC-370284-mediated inhibitory effect on AML cell viability/growth, and thereby confer drug resistance to the AML clones. To test this, we chose *JAK1*

and *MSH3* as two representatives and cloned constructs carrying the *JAK1*[*A893G*] mutant or the *MSH3*[*V600I*] mutant that was detected in our drug-resistant THP-1 cells (Supplementary Table 4). As expected, forced expression of either mutant at least partially reversed the inhibitory effect of NSC-370284 on AML cell viability (Supplementary Fig. 4c, d).

We next showed knockdown of *STAT3* and/or *STAT5* in MONOMAC-6 and THP-1 cells resulted in a downregulation of *TET1*, but not *TET2* or *TET3* (Fig. 3b; Supplementary Fig. 4e–h). Through searching the UCSC Genome Browser (https://genome.ucsc.edu/index.html), we found that STAT5 has a putative binding site (ttccctgaacagcttttaca tgtg; the consensus binding motif: ttcnnngaa; Fig. 3c, Site 4) located within the promoter region of *TET1* gene, suggesting *TET1* might be a direct target of the STAT proteins. The direct binding of STAT3 and STAT5 on the *TET1* loci was further validated in MONOMAC-6 cells through chromatin immunoprecipitation (ChIP)-qPCR assay, and such binding could be disturbed by NSC-370284 treatment (Fig. 3c–g). In NB4 cells, no significant binding of STAT3 or STAT5 on the *TET1* loci was detected (Fig. 3h). JAK1 was known

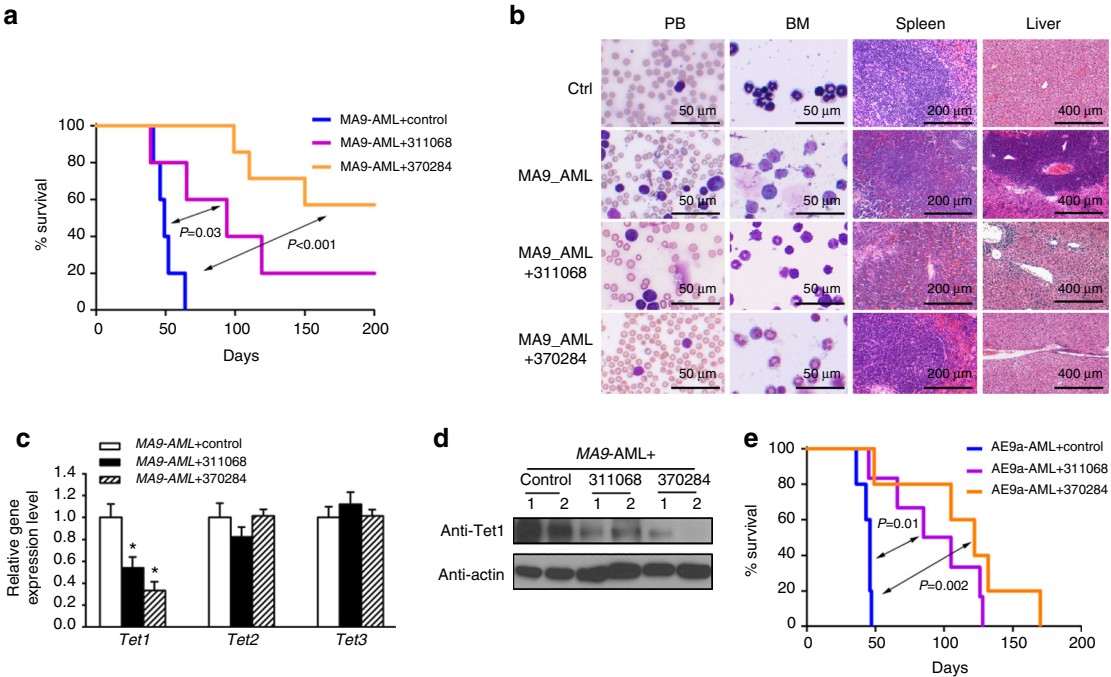

**Fig. 2** Therapeutic effects of NSC-311068 and NSC-370284 in AML in vivo. **a** NSC-311068 or NSC-370284 administration inhibits *MLL-AF9*-AML. The secondary BMT recipient mice were transplanted with leukemic BM blast cells collected from primary *MLL-AF9* AML mice. Upon the onset of leukemia, the recipient mice were treated with DMSO (control; n = 5), 2.5 mg/kg NSC-311068 (n = 5) or NSC-370284 (n = 7), i.p., once per day, for 10 days. Kaplan–Meier curves are shown. The *P* values were determined by log-rank test. **b** Wright–Giemsa staining of mouse peripheral blood (PB) and BM, or hematoxylin and eosin (H&E) staining of mouse spleen and liver of the treated or control leukemic mice. **c**, **d** Tet1/2/3 gene expression levels (**c**), or Tet1 protein level (**d**), in BM blast cells of the treated or control leukemic mice. *P < 0.05, two-tailed *t*-test. Error bar indicates SD of triplicate experiments. **e** Effects of NSC-311068 and NSC-370284 on the *AE9a*-AML mouse model. The secondary BMT recipient mice were transplanted with leukemic BM blast cells collected from the primary *AE9a*-AML mice. Upon the onset of leukemia, the recipient mice were treated with DMSO (control; n = 5), 2.5 mg/kg NSC-311068 (n = 6) or NSC-370284 (n = 6), i.p., once per day, for 10 days. Kaplan–Meier curves are shown. The *P* values were determined by log-rank test

as the upstream activator of the STAT pathway. To our surprise, we found knockdown of *TET1* resulted in a reduction of *JAK1* transcription in AML cells (Supplementary Fig. 4i). ChIP-qPCR results showed direct binding of TET1 to the *JAK1* promoter (Supplementary Fig. 4j). The above findings suggest JAK/STAT pathway promotes *TET1* transcription via direct binding of STAT3/5 to the *TET1* promoter, and TET1 also binds to the *JAK1* promoter and activates *JAK1* transcription. Herein we unveil a feedback loop between JAK1/STAT/TET1 in AML.

Our structural analysis suggested a potential direct binding of NSC-370284 to the conserved DNA-binding domain (DBD) of STAT3 or STAT5 (Fig. 3i). Such binding and the binding sites were identified by use of NMR chemical shift perturbation (CSP) [36]. Complex formation with compound NSC-370284 induced extensive CSPs at the isoleucine (Ile) residues of STAT3 at 1:2 of protein:ligand molar ratio (Fig. 3j). The CSPs occurred at residues adjacent to the DNA-binding site (I464) and those at or near DBD (Fig. 3j), indicating that compound NSC-370284 binds to STAT3 at or near DBD. The association between STAT3 and the *TET1* CpG island was further verified with electrophoretic mobility shift assay (EMSA); this association could almost be completely blocked by NSC-370284 (Fig. 3k; Supplementary Fig. 13c). In order to test whether NSC-370284 also inhibits the phosphorylation of STAT3 and STAT5, we treated MONOMAC-6 cells with NSC-370284 for a series of time points. Western blotting shows no significant alterations of the levels of STAT3 and STAT5 phosphorylation (Supplementary Figs. 4k, 13d, e). It is likely NSC-370284 mainly competes against DNA for STAT binding, but not suppresses STAT activation. These results indicate that STAT3 and STAT5 are direct targets of NSC-370284, which can interfere with the binding of STAT3/5 to *TET1*

promoter region and thereby suppress the transcription of *TET1*. Similarly, a previous study also reported a compound C48, a structural analog of NSC-370284, can bind directly to the DNA-binding domain of STAT3 protein and lead to apoptosis and inhibition of tumor cell growth [37]. Notably, the inhibitory effects of NSC-370284 on the expression of other STAT5 target genes, such as *HIF2α*, *IL2RA* and *FRA2*, were not as obvious as that on *TET1* (Supplementary Fig. 4l). The basal enrichment of STAT5 on the promoter of *HIF2α*[38], *IL2RA*[39], or *FRA2*[40] was very low, as compared with that on the *TET1* promoter; and the interruption by NSC-370284 on such association was much less obvious (Supplementary Fig. 4m). The very weak, if any, basal affinity of STAT5 with most of its target genes' promoters without cytokine stimulation (e.g., IL-2, IL-3, or EPO) has been reported before [38–41]. Therefore, our results revealed a very strong enrichment of STAT3/5 on *TET1* promoter (Fig. 3e–g, k; Supplementary Fig. 4l, m). Moreover, our results indicate that different from typical STAT inhibitors that target STAT kinase activity, NSC-370284 may exert its function mainly through interfering with the binding of STAT protein to the DNA regions which have relatively higher basal affinity to STATs, like the *TET1* promoter, in AML cells.

**Enhanced therapeutic efficacy of NSC-370284 analog UC-514321.** Based on the structure of the initial lead compound NSC-370284, we searched for structural analogs using BioVia Pipeline Pilot (Version 8.5.0.200) against the University of Cincinnati Compound Library, a collection of approximately 360,000 compounds. The 30 structurally most similar compounds were selected to explore the structure activity relationships (SAR)

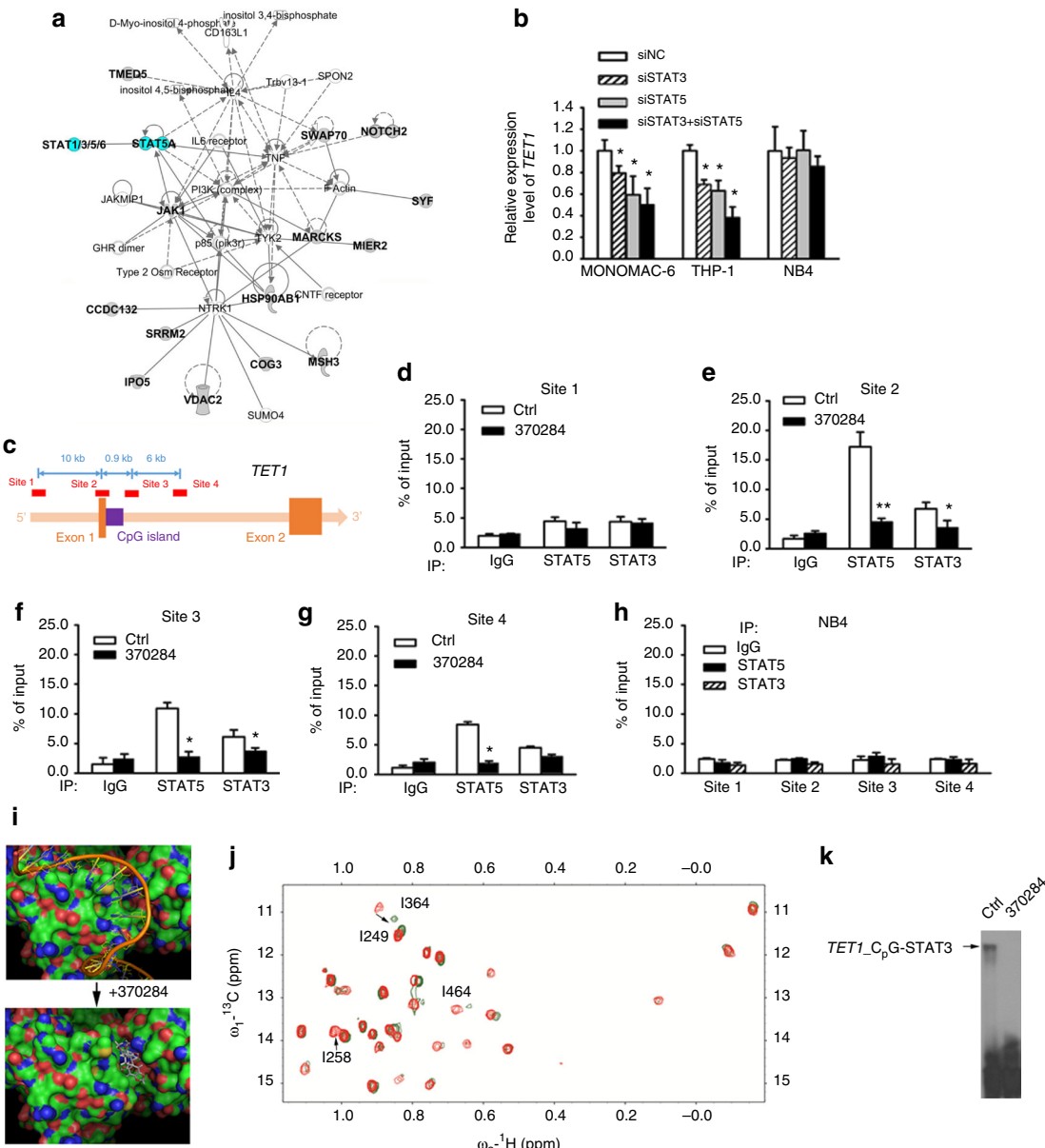

**Fig. 3** STAT3 and STAT5 are potential direct targets of NSC-370284 in AML. **a** A top network identified with IPA involving all the 14 genes carrying recurrent mutations in THP-1 NSC-370284-resistant clones. Genes with mutations are labeled in bold font. **b** Knockdown of *STAT3* and/or *STAT5* reduces *TET1* level. MONOMAC-6, THP-1, and NB4 cells were electroporated with control siRNA (siNC), si*STAT3*, si*STAT5* or a combination of si*STAT3* and si*STAT5*. *TET1* expression level was detected by qPCR 48 h post transfection. **c** Four genomic sites designed for ChIP-qPCR analysis to identify potential binding sites of STAT3 and STAT5 on *TET1* promoter and other regions. **d–h** MONOMAC-6 cells (**d–g**), were treated with DMSO control or 500 nM NSC-370284. ChIP-qPCR assay was carried out 48 h after drug treatment. Enrichment of STAT3, STAT5, or IgG at the *TET1* promoter region and other regions are shown. NB4 cells (**h**), were applied as a negative control. **i** Predicted binding of DNA (upper panel) or NSC-370284 (lower panel) with the DNA-binding domain (DBD) that is conserved between STAT3 and STAT5 proteins, from docking study on PDB ID 1bg1 using MolSoft ICM. **j** The association between STAT3 and NSC-370284 as determined with NMR chemical shift perturbation (CSP). Complex formation with compound 370284 induced extensive CSPs at the Ile residues of STAT3 at 1:2 of protein:ligand molar ratio (red peaks: free STAT3; green peaks: STAT3-NSC-370284 complex). The CSP occurs at residues adjacent to the DNA-binding site (I464), and residues at or near the DBD. **k** NSC-370284 suppresses the binding between STAT3 and *TET1* CpG island, as determined through EMSA. *$P < 0.05$; **$P < 0.01$, two-tailed $t$-test. Error bar indicates SD of triplicate experiments

(Supplementary Table 6). The majority of these compounds share the core aryl amine hydroxybenzodioxole scaffold with NSC-370284, varying primarily in the amine substituents and in the aryl substituents. Analogs lacking either the amine or hydroxyl were inactive. Results of MTS assays showed that one of the analog compounds, UC-514321, most significantly repressed MONOMAC-6 cell viability (Fig. 4a; Supplementary Fig. 5). UC-514321 showed an enhanced effect in repressing the viability of

*TET1*-high AML (including MONOMAC-6, THP-1, and KASUMI-1) cells, as compared with NSC-370284 and other JAK/STAT inhibitors, e.g. Pacritinib[42], KW-2449[43], STAT3/5 inhibitor Stattic[44], or STAT5 inhibitor sc-355979[45] (Fig. 4b–d). Similar to NSC-370284, UC-514321 showed no inhibitory effect on the viability of *TET1*-low AML (i.e., NB4) cells (Fig. 4e). Thus, the anti-tumor effect of UC-514321 is also TET1-signaling dependent. Notably, the STAT3/5 specific inhibitors Stattic and sc-

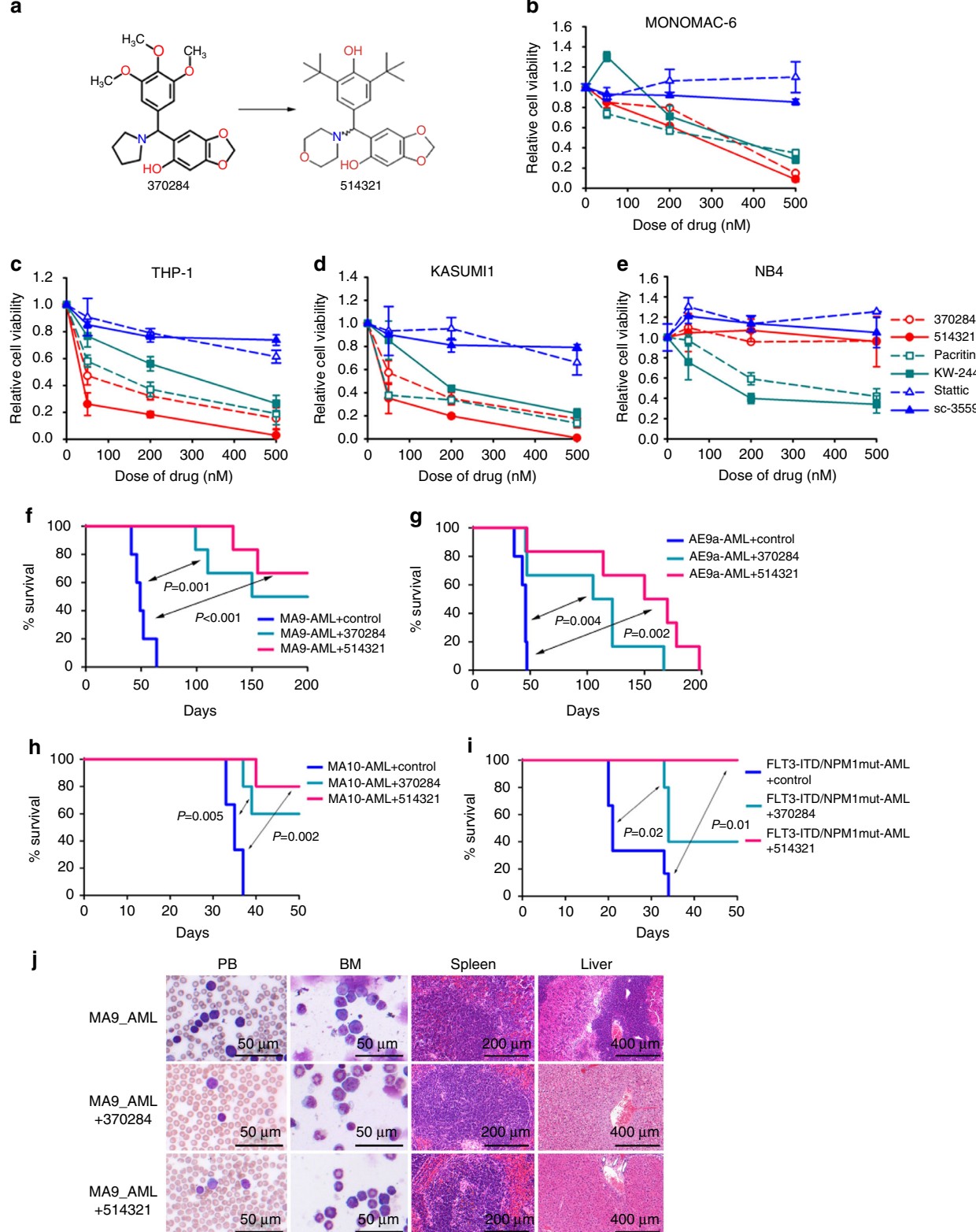

**Fig. 4** Effects of UC-514321 in treating AMLs. **a** Structures of NSC-370284 and UC-514321. **b–e** Effects of NSC-370284, UC-514321, and other JAK/STAT pathway inhibitors, i.e., Pacritinib, KW-2449, Stattic and sc-355979, on the viability of AML cell lines MONOMAC-6 (**b**), THP-1 (**c**), KASUMI1 (**d**), and NB4 (**e**). Cells were treated with drugs at indicated doses. Cell viability was detected by MTS 48 h post treatment. Error bar indicates SD of triplicate experiments. **f–j** Enhanced therapeutic effect of UC-514321, relative to NSC-370284, in treating *TET1*-high AMLs in vivo. Secondary BMT recipient mice were transplanted with primary leukemic BM cells with *MLL-AF9* (**f**), *AML-ETO9a* (**g**), *MLL-AF10* (**h**), or *FLT3*-ITD/*NPM1^mut^* (**i**). Upon the onset of leukemia, the recipient mice were treated with DMSO (control) (*n* = 5 or 6), 2.5 mg/kg NSC-370284 (*n* = 5 or 6), or UC-514321 (*n* = 5 or 6), i.p., once per day, for 10 days. Kaplan–Meier curves are shown. The *P* values were determined by log-rank test. **j** Wright–Giemsa staining of mouse PB and BM, or H&E staining of mouse spleen and liver of *MLL-AF9* AML secondary BMT recipients with or without drug treatment

355979 did not show significant inhibitory effects on the viability of these AML cells, probably because their EC$_{50}$ values are higher than 500 nM, the maximum concentration we tested. Overall, compared to other JAK and/or STAT inhibitors, UC-514321 is more effective and selective in inhibiting the viability of TET1-high AML cells.

Moreover, compared to the parental compound (NSC-370284), UC-514321 also showed an improved therapeutic effect in AML mouse models in vivo. In MLL-AF9-AML mice, UC-514321 prolonged median survival from 49 days (control) to > 200 days, better than NSC-370284 (Fig. 4f). In the AE9a-AML model, the median survival of UC-514321 treated mice was 160 days, much longer than the control (46 days) and NSC-370284 (114 days) treated groups (Fig. 4g). Thus, in both AML animal models, UC-514321 prolonged the median survival over three fold (Fig. 4f, g). Moreover, we tested the therapeutic effects of NSC-370284 and UC-514321 in two other AML models, i.e., MLL-AF10 AML and FLT3-ITD/NPM1$^{mut}$ AML[46]. NSC-370284 prolonged median survival of MLL-AF10 leukemic mice from 35 days to >50 days, and that of FLT3-ITD/NPM1$^{mut}$ leukemic mice from 21 days to 34 days. UC-514321 showed an even better therapeutic effect, as it prolonged median survival of both MLL-AF10 and FLT3-ITD/NPM1$^{mut}$ leukemic mice to >50 days (Fig. 4h, i). Notably, UC-514321 treatment cured 66.7% (4 out of 6) of the MLL-AF9 AML mice (Fig. 4f, j), and none of the UC-514321-treated FLT3-ITD/NPM1$^{mut}$ AML recipients developed full-blown AML within 50 days (Fig. 4i).

To determine whether the mechanism underlying UC-514321 function is similar to that of NSC-370284, we performed a series of mechanistic studies. As expected, similar to NSC-370284, UC-514321 significantly repressed expression of TET1, but not TET2 or TET3, along with the downregulation of putative target genes of TET1, e.g., HOXA7, HOXA10, MEIS1, PBX3, FLT3[19,47], etc., and the upregulation of negative targets, e.g., miR-22[46], in MONOMAC-6 cells (Fig. 5a, b). Notably, compared to NSC-370284, UC-514321 exhibited an enhanced effect on the changes in gene expressions (Fig. 5a, b). Both NSC-370284 and UC-514321 significantly reduced global 5hmC levels (Fig. 5c; Supplementary Fig. 13f, g), and their inhibitory effect was further confirmed with genome-wide 5hmC-seq in AML cells. We found both NSC-370284 and UC-514321 treatments resulted in around 75% reduction of 5hmC peak enrichment in the whole genome (Supplementary Fig. 6). Similar to NSC-370284 (Fig. 3j), the largest CSPs induced by UC-514321 treatment were also observed at the residues adjacent to DNA-binding surface (I431) and those near or in DBD (Fig. 5d). In contrast, Stattic and sc-355979 target the STAT SH2 domain[44,45]. Therefore, both NSC-370284 and UC-514321 induce CSP at residues at or adjacent to the DNA-binding surface, and thus they likely interfere with STAT protein's binding to DNA. And this was further verified with ChIP-qPCR assays. In MONOMAC-6 cells, UC-514321 treatment also resulted in a remarkable repression on the binding of STAT3/5 to TET1 promoter, even more significant than NSC-370284 at the CpG region (Fig. 5e–h).

In wild-type mouse BM progenitor cells transduced with MLL-AF9, treatment with NSC-370284 or UC-514321 resulted in a remarkable repression of cell viability, and no such inhibition was observed in Tet1-deficient counterpart cells (Fig. 5i–l). Other JAK/STAT inhibitors (i.e. Pacritinib, KW-2449, Stattic, and sc-355979) showed no significant differences in affecting cell viability between wild-type and Tet1-deficient cells (Fig. 5i, j; Supplementary Fig. 7a–d). Further, in MLL-AF9-transduced BM progenitor cells from conditional Tet1 knockout mice, NSC-370284 and UC-514321 also showed no significant effect in inhibiting the viability of the cells with induced Tet1 deficiency (Fig. 5m; Supplementary Fig. 7e). In addition, we created a

pLenti-puro vectored Tet1 construct, and transduced it into THP-1 and MONOMAC-6 cells and showed that ectopic expression of Tet1 sufficiently reversed the inhibitory effects of NSC-370284 and UC-514321 and restored viability of the AML cells (Fig. 5n, o). Taken together with the selective effect of NSC-370284 on TET1-high AML cell lines (Fig. 1b), the above results suggest that the anti-leukemic effects of NSC-370284 and UC-514321 are TET1-signaling dependent.

**Toxicity profiling of NSC-370284 and UC-514321.** Cell viability and apoptosis assays were firstly carried out to assess the effects of NSC-370284 and UC-514321 on normal hematopoietic stem/progenitor cells (HSPCs; herein we used c-Kit+BM cells) in vitro. Remarkably, NSC-370284 or UC-514321 treatment dramatically suppressed the viability of AML cells, but not that of normal HSPCs (Supplementary Fig. 8a). In addition, the compounds significantly increased apoptosis in AML cells, but not in normal HSPCs (Supplementary Fig. 8b). Thus, these two compounds showed no obvious toxicity on normal HSPCs. This is consistent with endogenous Tet1 expression pattern, as AML cells with MLL-AF10 or FLT3-ITD/NPM1$^{mut}$ have relatively higher Tet1 expression levels, as compared with normal HSPCs (Supplementary Fig. 8c). To assess potential toxicity of NSC-370284 and UC-514321 in normal tissues, especially the hematopoietic system, in vivo, we injected NSC-370284 or UC-514321 into normal C57BL/6 mice and assessed potential acute toxicity (24 h) or long-term (200 days) toxicity after 10 succeeding days' administration of the drug. We assessed body weights, spleen and liver weights, white blood cell (WBC) counts, all peripheral blood lineages, as well as granulocytes (Mac1$^{+}$Gr1$^{+}$), monocytes (Mac1$^{+}$Gr1$^{-}$) and progenitor (c-Kit$^{+}$) lineages of BM cells, and observed no evidence of either acute or long-term toxicity (Supplementary Figs. 9,10; Supplementary Table 7). The maximum tolerated dose of NSC-370284 and UC-514321 in mice was 65–85.6 mg/kg (Supplementary Table 8). The LD50 was around 123 mg/kg (Supplementary Table 8). Analysis of the pharmacokinetic (PK) properties of UC-514321 showed that the compound had a half-life of 11.02 h in mouse blood (Supplementary Table 9; Supplementary Fig. 11).

**Synergistic effect with standard chemotherapy.** Long-term treatment with a drug may cause drug resistance in patients, and thus combinatory therapy is often required. To treat AML cells that have gained resistance to our inhibitors, we investigated the effects of a set of first-line AML chemotherapy drugs including daunorubicin (DNR), cytarabine (AraC), all-trans retinoic acid (ATRA), azacytidine (AZA), and decitabine (DAC) on the viability of parental THP-1 cells and three NSC-370284-resistant clones (Fig. 6a; Supplementary Fig. 12a–d). Strikingly, all three drug-resistant clones appeared to be much more sensitive to DNR than the parental control (Fig. 6a). NSC-370284 or UC-514321 works synergistically with DNR on inhibiting the viability of THP-1 and KASUMI-1 cells (Fig. 6b, c). The synergistic effect between NSC-370284 or 514321 and the standard chemotherapy (i.e., the "5+3" regimen[48]) was further validated in vivo. Even administrated at relative low doses, the combinatorial treatment of NSC-370284+DNR/AraC or UC-514321+DNR/AraC showed a better therapeutic effect in curing MLL-AF9 AML, as the combinatorial administrations cured 83.3% (5 out of 6) of the AML mice (Fig. 6d, e). Through analysis of the RNA-seq data of the NSC-370284-resistant clones and parental cells, we showed that several gene clusters that are known to be associated with drug response, especially response to topoisomerase II inhibitors such as DNR, are enriched in NSC-370284-resistant cells. These gene clusters include JAK/STAT signaling, G2M checkpoint,

MYC targets, E2F targets, etc. (Supplementary Table 10a). A potential DNR sensitizing mechanism might be through targeting G2M checkpoint. It was shown that overexpression of *CDC25*, a key phosphatase of G2M checkpoint control, could significantly sensitize tumor cells to doxorubicin treatment[49]. Our RNA-seq data showed increased *CDC25* levels in NSC-370284-resistant

AML clones, relative to the parental cells. Also consistent is the enrichment of G2M checkpoint gene cluster in control samples relative to NSC-370284-treated samples (Supplementary Table 10b). The activation of JAK/STAT pathway might, directly or indirectly, contribute to G2M checkpoint abnormality, as reported previously by others[50,51]. Therefore, very likely the

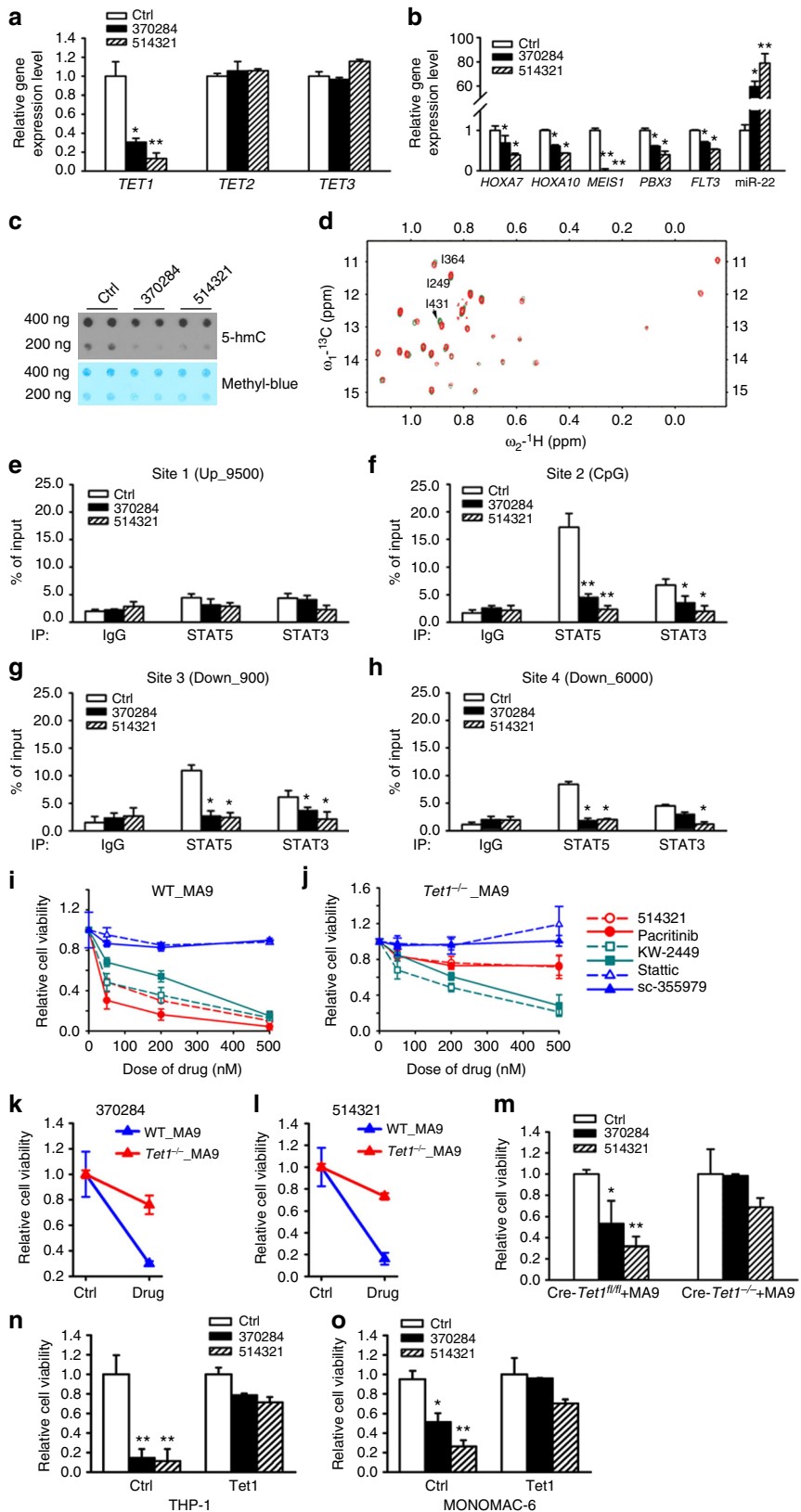

deregulation of G2M checkpoint in NSC-370284-resistant clones at least partially explains why the resistant clones, compared to the parental AML cells, are more sensitive to DNR treatment. Overall, the combination of NSC-370284 or UC-514321 with DNR represents a promising therapeutic strategy that will not only be effective in treating patients with *TET1*-high AMLs at relative low doses, but also avoid the occurrence of resistance to NSC-370284 or UC-514321.

## Discussion

While knockout of *Tet1* expression shows only very minor effects on normal development including hematopoiesis[21], our recent studies demonstrated that TET1 plays a critical oncogenic role in AML through promoting expression of oncogenic targets (e.g., *HOXA9*, *MEIS1*, *PBX3*, etc.) and repressing expression of tumor-suppressor targets (e.g., miR-22)[18,19,46]. Thus, targeting TET1 signaling is a promising therapeutic strategy to treat *TET1*-high AMLs. In order to target critical oncogenic proteins with catalytic activity, one of the most popular approaches is to interfere with the catalytic activity of oncogenic proteins, such as FLT3 inhibitor Quizartinib that represses the kinase activity of FLT3[52], STAT inhibitor Stattic that blocks the dimerization of STAT3[44], etc. However, as shown in a number of clinical reports, treatments of catalytic activity inhibitors often result in aberrant upregulation of the target oncoproteins or trigger gene mutations, which eventually leads to drug resistance[53,54]. The discovery of the bromodomain and extra-terminal (BET) inhibitor JQ1 as an effective strategy to target c-Myc signaling[52,55] suggested an alternative strategy of repressing the TET1 signaling instead of directly targeting the enzymatic activity of TET1. Any drugs that efficiently target the expression, i.e., transcription, translation, or degradation, of the oncogenes or oncogenic proteins could largely avoid drug resistance caused by target oncogene upregulation or constitutively activated gene mutations. Moreover, we reported previously that TET1 can recruit polycomb proteins to the promoter region of the mir-22 gene and suppress the primary transcription of this critical tumor-suppressor microRNA, and such transcriptional suppression is independent from TET1's enzymatic activity[46]. Therefore, instead of seeking inhibitors targeting TET1 enzymatic activity directly that are unable to fully repress the function of TET1, we screened for inhibitors that suppress *TET1* expression in this study for treating AML.

Through correlation analysis of cell response to 20,602 chemical compounds and *TET1* levels of in the NCI-60 collection of cancer cell samples, followed by MTS assays of top drug candidates in AML cells, we finally narrowed down to two candidate chemical compounds (i.e., NSC-311068 and NSC-370284) that both suppressed AML cell viability and *TET1* expression. Importantly, both NSC-311068 and especially NSC-370284 showed remarkable therapeutic effects in curing AML in vivo.

UC-514321, a structural analog of NSC-370284, exhibited a more potent anti-leukemic activity in vitro and in vivo than NSC-370284. Mechanistically, our TET1-signaling inhibitors directly target STAT3/5, which are direct upstream regulators of *TET1* transcription. Remarkably, compared to currently available JAK/STAT inhibitors (e.g., Pacritinib, KW-2449, Stattic, and sc-355979), our compounds (NSC-370284 and UC-514321) exhibit a much higher selectivity and also a higher efficacy in targeting *TET1*-high AML, which likely due to the unique property of our TET1-signaling inhibitor as they directly bind to the DBD of STAT3/5, interfere with the binding of STAT3/5 to *TET1* promoter region, and thereby repress the transcription of *TET1*. Moreover, we show that both NSC-370284 and UC-514321 exhibit a synergistic effect with DNR in treating *TET1*-high AML cells in vitro and in vivo. Notably, NSC-370284-resistant THP-1 AML cells are even more sensitive to DNR than parental THP-1 cells. Taken together, our findings highlight the therapeutic potential of targeting TET1, a key oncogenic epigenetic regulator related to DNA demethylation, in AML. Our data also reveal that STAT3 and STAT5 are direct upstream regulators of *TET1*, and they are druggable targets to suppress TET1 signaling. Our data suggest that application of small-molecule compounds (e.g., NSC-370284 and UC-514321) that selectively and effectively target the STAT/TET1 signaling, especially in combination with standard chemotherapy agents, represents an effective novel therapeutic strategy for the treatment of *TET1*-high AML (including *MLL*-rearranged AML and t(8;21) AML), which accounts for approximately 30% of total AML cases. Moreover, these effective inhibitors can also be employed as tool compounds in both basic and translational research to selectively target the STAT/TET1 signaling axis and suppress 5hmC globally.

## Methods

**Animal studies**. C57BL/6 (CD45.2) and B6.SJL (CD45.1) mice were purchased from the Jackson Lab (Bar Harbor, ME, USA) or Harlan Laboratories, Inc. (Indianapolis, IN, USA) and maintained in house. Both male and female mice were used for the experiments. All laboratory mice were maintained in the animal facility at University of Cincinnati or University of Chicago. All experiments on mice in our research protocol were approved by Institutional Animal Care and Use Committee (IACUC) of University of Cincinnati or University of Chicago. All methods were performed in accordance with the relevant guidelines and regulations. The maintenance, monitoring, and end-point treatment of mice were conducted as described previously[46,56,57]. Randomization, allocation concealment, and blind outcome assessment were conducted throughout all the experiments.

**Mouse bone marrow transplantation and drug treatment**. Secondary mouse BMT was carried out as described previously[46,56,57]. Upon the onset of leukemia (when mice had an engraftment (CD45.1) of over 20% and/or white blood cell counts higher than $4 \times 10^9$/L, usually 10 days post transplantation), the recipient mice were injected with DMSO control, 2.5 mg/kg NSC-311068, NSC-370284, or UC-514321, i.p., once per day, for 10 days. For the "5+3" and NSC-370284 or UC-514321 combination treatment experiment, after the onset of AML, the recipient mice were treated with PBS control, or NSC-370284 or UC-514321 alone, i.p., once per day, for 10 days or together with the "5+3" treatment[48]. For the "5+3"

---

**Fig. 5** NSC-370284 and UC-514321 function as *TET1*-transcription inhibitors in *TET1*-high AMLs and their anti-leukemic effects are TET1-dependent. **a** NSC-370284 and UC-514321 inhibit the transcription of *TET1*, but not *TET2* or *TET3*. MONOMAC-6 cells were treated with DMSO control, 500 nM NSC-370284 or UC-514321 for 48 h. Gene expression levels are shown. **b** Effects of NSC-370284 and UC-514321 in downstream gene targets of TET1. **c** NSC-370284 and UC-514321 repressed global 5hmC level in MONOMAC-6 cells. **d** The association between STAT3 and UC-514321 as determined with NMR CSPs (red peaks: free STAT3; green peaks: STAT3-UC-514321 complex). **e–h** MONOMAC-6 cells were treated with DMSO control, 500 nM NSC-370284 or UC-514321. ChIP-qPCR assay was carried out 48 h after drug treatment. Enrichment of STAT3, STAT5, or IgG at the *TET1* promoter region and other regions are shown. **i–l** Functions of NSC-370284 and UC-514321 depend on *Tet1* expression. BM progenitor cells of wild-type or *Tet1*$^{-/-}$ mice were retrovirally transduced with *MLL-AF9*. Infected cells were treated with NSC-370284, UC-514321, and other JAK/STAT pathway inhibitors, i.e., Pacritinib, KW-2449, Stattic and sc-355979, at indicated doses (**i**, **j**), or at a particular dose (i.e., 500 nM); **k**, **l** for 48 h. Relative cell viabilities are shown. **m** Cre-*Tet1*$^{fl/fl}$ mouse BM progenitor cells were retrovirally transduced with *MLL-AF9*. Transduced cells were induced with polyI:C for 7 days, and then treated with 500 nM NSC-370284, UC-514321, or DMSO control for 48 h. Relative cell viability is shown. **n–o** THP-1 (**n**), and MONOMAC-6 (**o**), cells were infected with lentivirus of pLenti-puro vector-based *Tet1* construct. Cells with or without doxycyclin inducing were treated with 250 nM NSC-370284, UC-514321 or DMSO control for 24 h. Relative cell viability is shown. *$P < 0.05$; **$P < 0.01$, two-tailed *t*-test. Error bar indicates SD of triplicate experiments

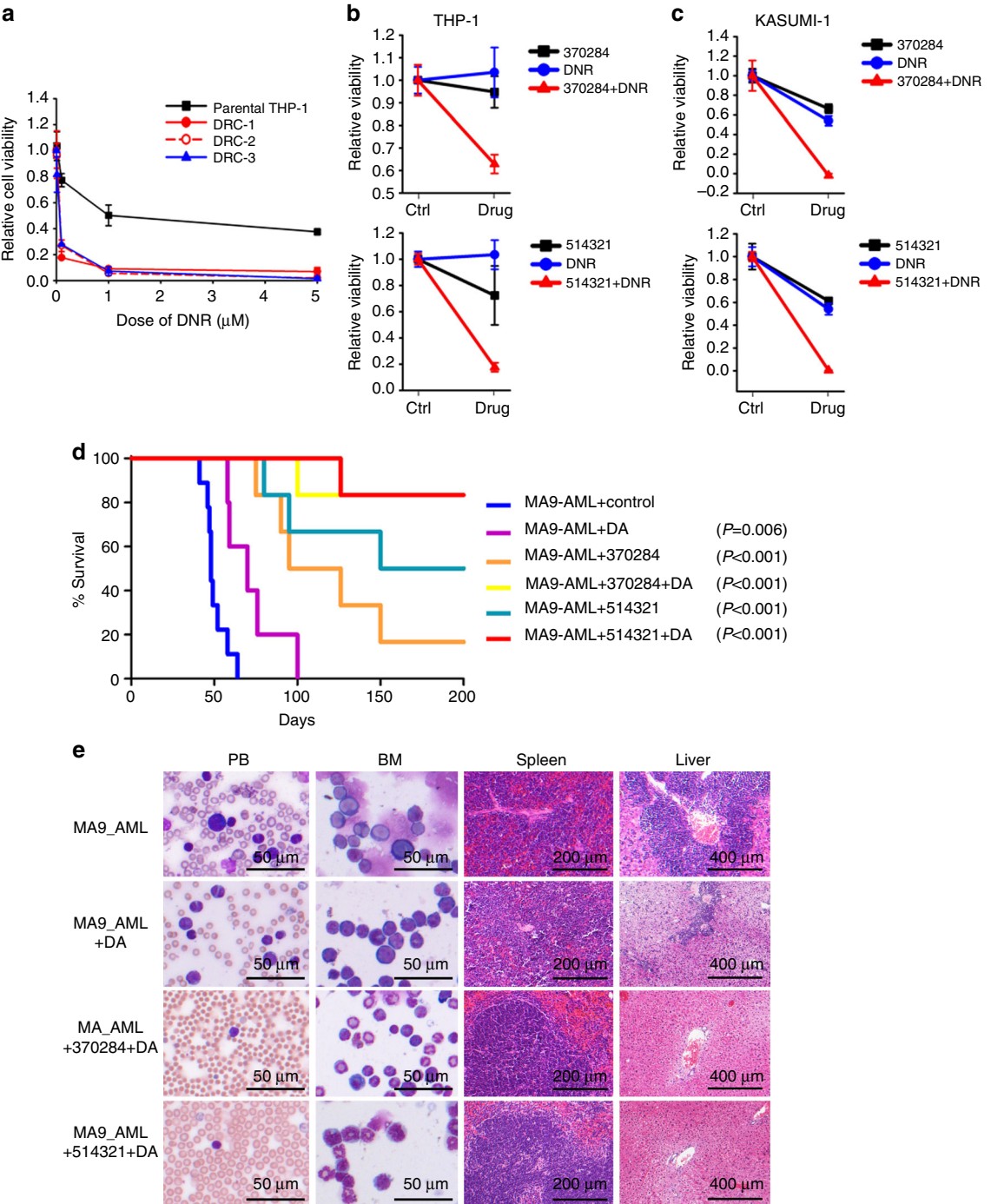

**Fig. 6** Synergistic effect of NSC-370284 or UC-514321 with daunorubicin in treating *TET1*-high AMLs in vitro and in vivo. **a** NSC-370284-resistant AML cells are sensitive to DNR. Three of the THP-1 NSC-370284-resistant clones (DRC-1–3) and the parental control were treated with DNR at indicated doses. Cell viability was tested 48 h after the treatments. **b**, **c** THP-1 (**b**), and Kasumi-1 (**c**), cells were treated with DMSO (Ctrl), 25 nM NSC-370284 (upper panels) or UC-514321 (lower panels), and/or 100 nM DNR. Relative cell viability at 48 h post drug treatment is shown. Error bar indicates SD of triplicate experiments. **d** Synergistic therapy of NSC-370284 or UC-514321 in combination with standard chemotherapy. Secondary BMT recipient mice were transplanted with primary *MLL-AF9* leukemic BM cells. Upon the onset of leukemia, the mice were treated with DMSO (control), "5+3" regimen[48] alone (i.e., a daily dose of 50 mg/kg Ara-C for five days along with a daily dose of 3 mg/kg DNR during the first three days of Ara-C treatment), or in combination with 10 days' NSC-370284 or UC-514321 treatment, 2.5 mg/kg, i.p., once per day. Five to nine mice were included in each group. Kaplan–Meier curves are shown. The *P* values were determined by log-rank test. **e** Wright–Giemsa staining of mouse PB and BM, or H&E staining of mouse spleen and liver of *MLL-AF9* secondary BMT recipients treated with "5+3" alone, or combinational therapy

treatment, AraC (Cytarabine, Bedford Laboratories) and DNR (Daunorubicin, Sigma-Aldrich) were reconstituted with PBS, filtered, and stored in aliquots at −20 ºC. The "5+3" therapy regimen consists of five consecutive daily doses of 50 mg/kg AraC along with 3 mg/kg DNR daily during the first three days of treatment. Drugs were delivered by tail vein and intraperitoneal injection. Weights were taken daily during treatment and doses were recalculated to ensure the mice received a consistent dose of 50 mg/kg AraC and 3 mg/kg DNR every treatment.

**In vivo preclinical pharmacokinetic analysis**. PK studies of UC-514321 were performed in C57BL/6 mice (18–22 g). Prior to the study, mice were fasted for 12 h with free access to water. Animals were housed on a 12-h light/dark cycle at 22–24 °C and 30–50% relative humidity. 15.00 mg/kg UC-514321 was administered through intraperitoneal (i.p.) injection. After dosing, blood samples were collected at indicated time points (15 min, 30 min, 1 h, 2 h, 4 h, 8 h, 12 h, and 24 h). Samples from three animals were collected at each time point. Approximately 500 µL of blood was collected via orbital vein from each mouse anesthetized with isoflurane. In total, 200 µL blood was then mixed with 600 µL methanol by vortex, spun twice at 11,000 rpm, 4 °C, for 15 min to remove insoluble matter. Supernatant was collected and transferred to 2 mL liquid sample vials. An aliquot of 40 µL mixture solution was injected for LC analysis. Standard curves were prepared in blood covering the concentration range of 50–3,0000 ng/mL. Using the data from the standard curves, calibration curves were generated for PK tests. The PK parameters were calculated using a noncompartmental model with PKsolver[58].

**Cell culture and drug treatment**. MONOMAC-6, THP-1, KOCL-48, KASUMI-1, ML-2, and NB4 cells were purchased from ATCC (Manassas, VA), and cultured as described previously[57,59]. All cell lines were tested for mycoplasma contamination yearly using a PCR Mycoplasma Test Kit (PromoKine) and were proven to be mycoplasma negative. All cell lines were authenticated through STR profiling yearly.

**Cell transfection and retrovirus infection**. siRNAs were transfected into MONOMAC-6 cells with Cell Line Nucleofector Kit V following program T-037, using the Amaxa Nucleofector Technology (Amaxa Biosystems, Berlin, Germany). Experiments were performed 48 h after transfection. Retrovirus infection of mouse BM progenitor cells were conducted as described previously with some modifications[57,60]. Briefly, retrovirus vectors were co-transfected with pCL-Eco packaging vector (IMGENEX, San Diego, CA) into HEK293T cells using Effectene Transfection Reagent (Qiagen, Valencia, CA) to produce retrovirus. BM cells were harvested from a cohort of 4- to 6-week-old C57BL/6, $Tet1^{-/-}$, or Cre-$Tet1^{fl/fl}$ donor mice after 5 days of 5-fluorouracil (5-FU) treatment, and primitive hematopoietic progenitor cells were enriched with Mouse Lineage Cell Depletion Kit (Miltenyi Biotec Inc., Auburn, CA). An aliquot of enriched hematopoietic progenitor cells was added to retroviral supernatant together with polybrene in cell culture plates, which were centrifuged at $2000 \times g$ for 2 h at 32 °C (i.e., "spinoculation") and then the medium was replaced with fresh media and incubated for 20 h at 37 °C. Next day, the same procedure was repeated once. Infected cells were grown in RPMI 1640 medium containing 10 ng/mL murine recombinant IL-3, 10 ng/mL IL-6, 100 ng/mL murine recombinant SCF (R&D Systems, Minneapolis, MN), along with 1.0 mg/ml of G418. Experiments were performed 7 days after infection.

**RNA extraction and quantitative RT-PCR**. Total RNA was extracted with the miRNeasy extraction kit (Qiagen) and was used as template for quantitative RT-PCR (qPCR) analysis as described previously[19,46,57].

**Cell apoptosis and proliferation assays**. These experiments were conducted as described previously[57,59] with ApoLive-Glo Multiplex Assay Kit, or CellTiter 96 AQ$_{ueous}$ Non-Radioactive Cell Proliferation Assay Kit (Promega, Madison, WI).

**NMR chemical shift perturbation**. Specific Ile-methyl labeled STAT3 for NMR studies was prepared as described previously[36]. For each compound, STAT3 was expressed and purified fresh, and a reference spectrum was acquired. Then, the complexes of STAT3 with each of the compounds in the same buffer as the free STAT3 (reference) sample were prepared and two-dimensional (2D) $^1$H-$^{13}$C-HMQC spectra of STAT3 were acquired. The protein samples contained 20 µM STAT3. Both compounds were added to a final concentration of 40 µM, respectively. All 2D $^1$H-$^{13}$C HMQC spectra were collected with $2048 \times 128$ complex points at 35 °C on the Bruker Ascend 700 spectrometer equipped with a cryoprobe. The spectra were analyzed with the program Sparky (T.D. Goddard and D.G. Kneller, SPARKY 3, University of California, San Francisco).

**Chromatin immunoprecipitation-qPCR**. ChIP assay was conducted as described previously[50], with SABiosciences Corporation's ChampionChIP One-Day Kit (Qiagen, Frederick, MD) following the manufacturer's protocol. Chromatin from MONOMAC-6 cells were cross-linked, sonicated into an average size of ~500 bp, and then immunoprecipitated with antibodies against STAT3 (C-20, Santa Cruz Biotechnology, Dallas, TX), STAT5 (610191, BD Biosciences, San Jose, CA), TET1

(Y-14, Santa Cruz, Dallas, TX), or IgG (ab2410, Abcam, Cambridge, MA). Purified DNA was amplified by real-time qPCR using primers targeting the promoter of *TET1* as described before[19]. Sequences of qPCR primers for the TET1 promoter sites are: Site 1 forward: 5′-ACTTTGACCTCCCAAAGTGCTGGA-3′, reverse: 5′-ACCTGAGTGATGCTGAGACTTCCT-3′; Site 2 forward: 5′-TTTGGGAACC-GACTCCTCACCT-3′, reverse: 5′-TCGGGCAAACTTTCCAACTCGC-3′; Site 3 forward: 5′-ACGCTGGGCATTTCTGATCCACTA-3′, reverse: 5′-TATTGTG-CAGCTCGTTTAGTGCCC-3′; Site 4 forward: 5′-CCATCTCCCGACACACA-3′; reverse: 5′-TTGGCAGTGACCTTGAGA-3′.

**Electrophoretic-mobility shift assay**. EMSA was conducted with EMSA Assay Kit (Signosis, Santa Clara, CA) according to the manufacturer's protocol with minor modifications. Briefly, purified STAT3 protein was incubated with Biotin-labeled *TET1*-CPG probe (hot probe) and/or cold probe, and then protein/DNA complexes were separated on a non-denaturing polyacrylamide gel. Bands were detected using Streptavidin-HRP conjugate and a chemiluminescent substrate. The sequences of the *TET1*-CPG probe are: Forward: 5′ Biotin-CCGGTAGGCGTCCTCCGCGACCCGC-3′; Reverse: 5′ Biotin-GCGGGTCGCGGAGGACGCCTACCGG-3′.

**Western blotting**. Cells were washed twice with ice-cold phosphate-buffered saline (PBS) and ruptured with RIPA buffer (Pierce, Rockford, IL) containing 5 mM EDTA, PMSF, cocktail inhibitor, and phosphatase inhibitor cocktail. Cell extracts were microcentrifuged for 20 min at $10,000 \times g$ and supernatants were collected. Cell lysates were resolved by SDS-PAGE and transferred onto PVDF membranes. Membranes were blocked for 1 h with 5% skim milk in Tris-buffered saline containing 0.1% Tween 20 and incubated overnight at 4 °C with anti-Tet1 (1:1000) (GT1462, GeneTex, Irvine, CA), anti-STAT3 (1:1000) (C-20, Santa Cruz Biotechnology, Dallas, TX), anti-STAT3 (phosphor-Y705) (1:1000) (ab76315, Abcam, Cambridge, UK), anti-STAT5 (1:1000) (610191, BD Biosciences, San Jose, CA), anti-STAT5 (phospho-Tyr694) (1:1000) (9351S, Cell Signalling Technology Inc., Danvers, MA), anti-JAK1 (1:1000) (ab133666, Abcam), anti-JAK1 (phosphor-Y1022/Y1023) (1:1000) (ab138005, Abcam), anti-GAPDH (1:1000) (sc-47724, Santa Cruz Biotechnology), or anti-ACTIN antibody (1:1000) (8H10D10, Cell Signaling Technology Inc.). Membranes were washed 30 min with Tris-buffered saline containing 0.1% Tween-20, incubated for 1 h with appropriate secondary antibodies conjugated to horseradish peroxidase, and developed using chemiluminescence substrates.

**5hmC labeling reaction and dot blotting**. The 5-hydroxymethylcytosine (5hmC) labeling reactions and 5hmC dot blotting were performed as described previously[19]. Briefly, 3 µg sonicated genomic DNA (100–500 bp) fragments were incubated in solution containing 50 mM HEPES buffer (pH 7.9), 25 mM MgCl$_2$, 100 µM UDP-6-N3-Glc, and 1 µM beta-glucosyltransferase (β-GT) for 1 h at 37 °C. The CLICK was performed with addition of 150 µM dibenzocyclooctyne modified biotin into the purified DNA solution, and the reaction mixture was incubated for 2 h at 37 °C. Six hundred nanograms of labeled genomic DNA samples were spotted on an Amersham Hybond-N+membrane (GE Healthcare, Little Chalfont, UK). DNA was fixed to the membrane by Stratagene UV Stratalinker 2400 (auto-crosslink). The membrane was then blocked with 5% BSA and incubated with Avidin-HRP (1:40,000) (Bio-Rad, Hercules, CA), and then visualized by enhanced chemiluminescence.

**5hmC-Seal library construction**. Hundred nanograms genomic DNA extracted from ML-2 cell were fragmented in 50 µL Tagmentation buffer at 55 °C. Fragmented DNA was purified by Zymo DNA clean&concentrator Kit (Zymo Research, Tustin, CA). Then, the selective 5hmC chemical labeling was performed in 25 µL glucosylation buffer (50 mM HEPES buffer pH 8.0, 25 mM MgCl$_2$) containing above fragmented DNA, 100 µM N$_3$-UDP-Glc, 1 µM β-GT, and incubated at 37 °C for 2 h. After purified in 45 µL ddH$_2$O, 1.5 µL DBCO-PEG4-Biotin (Click Chemistry Tools, 4.5 mM stored in DMSO) was added and incubated at 37 °C for 2 h. The biotin labeled DNA was pulled down by 5 µL C1 Streptavidin beads (Life Technologies, Carlsbad, CA) for 15 min at room temperature. Next, the captured DNA fragments were subjected to 13 cycles of PCR amplification using Nextera DNA sample preparation kit (Illumina, San Diego, CA). The resulting amplified product was purified by 1.0× AMPure XP beads. Input library was made by direct PCR from fragmented DNA without chemical labeling and capture. The libraries were quantified by a Qubit fluorometer (Life Technologies) and sequenced on an Illumina HiSEQ4000 sequencer with paired-end 50-bp reads.

**Mutation calling from RNA-Seq**. THP-1 cells were grown in RPMI1640 median containing 10% FBS and treated with 250 nM–1 µM NSC-370284 for 120 days. RNA was extracted using Qiagen miRNeasy kit. Library was prepared by PrepX mRNA Library kit (WaferGen) combined Apollo 324 NGS automated library prep system. Libraries at the final concentration of 15 pM were clustered onto a single read (SR) flow cell using Illumina TruSeq SR Cluster kit v3, and sequenced to 50 bp using TruSeq SBS kit on Illumina HiSeq system. RNA-Seq reads were aligned to the hg19 genome assembly using STAR with default parameters. GATK was used to call the mutations from mapped RNA-Seq reads. ANNOVAR was used to annotate

the mutations. The raw data were deposited to NCBI SRA under the accession number SRP103997.

**Statistical software and statistical analyses.** The gene network was analyzed with Ingenuity Pathway Analysis (Qiagen). The modeling of protein–DNA/chemical compound binding was conducted with Molsoft ICM-Pro (Molsoft L.L.C., San Diego, CA). The t-test, Kaplan–Meier method, log-rank test, etc. were performed with WinSTAT (R. Fitch Software), GraphPad Prism version 5.00 (GraphPad Software, San Diego, CA), and/or Partek Genomics Suite (Partek Inc.). The P values less than 0.05 were considered as statistically significant. For 5hmC sequencing analysis, illumina sequencing reads were mapped to UCSC hg19 human reference genome using bowtie program[61]. Only uniquely mapped reads were retained for the following data analysis. PCR duplicates were removed using samtools[62]. The identification of 5hmC peaks in each sample was performed using MACS[63], and an IDR cutoff of 0.01 was used to filter high confident peaks[64]. Peaks from different samples were merged together into a unified catalog of 5hmC enriched regions using HOMER[65]. To visualize sequencing signals in IGV[66], BigWig files were generated by deepTools[67] with RPKM normalization method. All the data meet the assumptions of the tests, with acceptable variation within each group, and similar variance between groups.

**Data availability.** Data referenced in this study are available in The Gene Expression Omnibus. The 5hmC sequencing data is available under GSE97407 (https://www.ncbi.nlm.nih.gov/geo/query/acc.cgi?token=etodaoikjtwnpgb&acc=GSE97407). RNA-Seq data is available under SRP103997 and GSE101480.

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

## Acknowledgements

The authors are grateful to Drs Junlin Guan, Saijuan Chen and Ruibao Ren for their constructive comments. We also thank Dr Xiang Zhang (supported by University of Cincinnati College of Medicine Core Enhancement Funding 2017) for the RNA-seq assays. This work was supported in part by the National Institutes of Health (NIH) R01 Grants CA211614 (J.C.), CA178454 (J.C.), CA214965 (J.C.), CA182528 (J.C.), RM1 HG008935 (C.H.), Gabrielle's Angel Foundation for Cancer Research (X.J.), The University of Chicago Committee on Cancer Biology (CCB) Fellowship Program (X.J.), National Natural Science Foundation of Chongqing (cstc2015jcyjBX0100) (C.L.), Foundation of Innovation Team for Basic and Clinical Research of Zhejiang Province (2011R50015) (J.J.). J.C. is a Leukemia & Lymphoma Society (LLS) Scholar. C.H. is an investigator of the Howard Hughes Medical Institute (HHMI). WCR and PPL are supported by intramural research programs of NCI and NHGRI, NIH, respectively.

## Author contributions

X.J. and J.C. conceived the project and designed the research. X.J., C.H., K.F., J.N., X.C., C.-H.C., L.C., Z.Z., C.H., Y.T., W.S., J.R.S., M.W., W.C.R., L.D., C.S., S.A., B.U., J.L., H. W., R.S., H.H., Y.W., C.L., X.Q., J.M., Y.Z., J.D., J.J., C.L., P.P.L., C.H., Y.C. and J.C. performed experiments and/or data analysis. X.J., X.C., Z.Z., M.W., J.L., J.M., Y.Z., J.D., C.L., W.S., C.H., Y.C. and J.C. contributed reagents/analytic tools and/or grant supports. X.J. and J.C. wrote the paper. All authors discussed the results and commented on the manuscript.

## Additional information

**Competing interests:** A patent (to X.J. and J.C.) was applied for based on this work. C.H. is a scientific founder of the Accent Therapeutics, Inc. The remaining authors declare no competing financial interests.

