## [Peer Review File · Nature Communications]

Reviewers' comments:

Reviewer #1 (Remarks to the Author):

TET1 is the founding member of a family of DNA methylcytosine dioxygenases (including TET1/2/3), which are known to convert 5-methylcytosine (5mC) to 5-hydroxymethylcytosine (5hmC), leading to DNA demethylation. While previous studies suggested that all three TET genes likely function as tumor-suppressor genes in various cancers, the Chen lab and others have provided compelling evidence showing that TET1 plays a critical oncogenic role in the development of myeloid malignancies. Importantly, Tet1 knockout has been reported to have minor effects on normal development including hematopoiesis in mice. Therefore, TET1 is a promising therapeutic target for AML treatment and targeting TET1 signaling is a hot pursuit in the field of cancer/myeloid malignancies. In this manuscript, Jiang et al. timely reported the identification of several novel therapeutic drugs against TET1 signaling. The authors demonstrated the high efficacy and selectivity of these drugs in targeting acute myeloid leukemias (AMLs) with high levels of TET1 expression (i.e., TET1-high AMLs) both in vitro and in vivo. To develop inhibitors targeting TET1 signaling, the authors employed an innovative strategy to screen small-molecule compounds that inhibit the transcription of TET1, rather than inhibiting TET1's enzymatic activity, since TET1 exerts its oncogenic role through both enzymatic-dependent and -independent mechanisms. Using this strategy, the authors identified compounds NSC-311068 and NSC-370284 as selective inhibitors of TET1 signaling by inhibiting TET1 transcription. They showed that both compounds, especially NSC-370284, exhibited potent therapeutic effects in treating TET1-high AMLs. A structural analogue of NSC-370284, namely UC-514321, exhibited even better therapy effects. In addition, the authors delineated the molecular mechanism by which these compounds inhibit the TET1 transcription, by showing that the compounds directly target the transcriptional activators of TET1, STAT3/5. The authors also showed that NSC-370284 or UC-514321 exhibit a synergistic therapeutic effect with standard chemotherapy agents such as daunorubicin. While, these compounds had no noticeable (both acute and long-term) side effects on normal hematopoietic system in vivo, highlighting the low toxicity of these compounds for future potential clinical application.

In summary, this is the first report for the identification of therapeutic agents targeting TET1 signaling, which exhibit high efficacy and selectivity for treating TET1-high AMLs. Moreover, it is novel to show that STAT3/5 are direct transcriptional regulators of TET1 and small-molecule inhibitors targeting STAT3/5 can efficiently suppress TET1 expression. This work is of great significance and novelty in both basic and clinical cancer research. The combination of such TET1 inhibitors with standard chemotherapy agents holds potential to improve the clinical treatment of TET1-high AMLs. I only have minor comments as the following:

Minor points:

1. While the therapeutic effects of the TET1 transcription inhibiting compounds are quite promising, the authors should provide more pre-clinical data such as PK/PD, which are important to evaluate the potential of clinical appliances.
2. Although the authors have shown that these TET1 transcription inhibiting compounds exhibited no obvious effects on normal hematopoiesis in vivo, it would be helpful to examine the effects of these compounds on normal HSC/HPCs in vitro, if possible compared with TET1-high HSC/HPCs. Colony-forming assay and cell viability (apoptosis) assay can be applied.

Reviewer #2 (Remarks to the Author):

Alteration in enzymatic DNA methylcytosine oxidation has been reported in AML. But the therapeutic opportunity has not been explored. This manuscript describes the first effort in

suppressing AML by inhibiting the function of TET oxidase which appears to act as an oncogene. The overall concept presented in this manuscript is novel.

More and more basic transcriptional machinery factors are now being explored for therapeutic applications with success and promises, such as BRD4 inhibitors (best example), integrator complex, TAF250 and so on. The inhibitors reported in this work actually targeted STATs but was identified by using TET1 as readout in a screen. If these compounds had reliable efficacy to AML cells in vivo and not much side effects, they would be promising in principle. However, the specificity to TET1 transcription inhibition has to be further studied. The question whether the compounds deliver their effects really through inhibiting TET1 transcription is not fully addressed. Targeting Stats would influence transcription of many genes. Additionally, AML is highly heterogenous in its genetics, therefore the mechanism that makes the compounds specifically effective for the AML with high TET1 expression would need to be better characterized.

A specific suggestion is to do inducible ectopic expression of TET1 in AML cells treated with UC-514321 and examine whether this would mitigate the effect of the compound, restoring cell viability.

The compounds appeared to work well in vivo with the MLL-AF9 and AML-ETO9a models. But how is their effect with other important models?

The title appears somewhat confusing as the compounds bind STAT3/5 as STAT inhibitors rather than inhibit TET1 transcription directly.

Reviewer #3 (Remarks to the Author):

In this manuscript, the authors focus on TET1 as a target in AML. Based on a chemical library screen database, they have identified several compounds that they suggest decrease TET1 expression in a STAT3- or STAT5-dependent manner, and exert a therapeutic effect in in vitro and murine models of AML.

The overall concept of this study is very interesting and provocative, and these compounds certainly do show activity in the experiments shown. However, the data leave open many fundamental questions regarding the mechanism by which these compounds are exerting their effect, and much of the data can be explained by non-specific toxicity, especially as at least one of the compounds is structurally related to a known topoisomerase II inhibitor. In particular, the following points should be considered:

1. The rationale for the strategy to identify the initial lead compounds should be clarified. In the NCI60 panel, only six of the lines are hematopoietic. Since TET1 may have a tumor suppressor role in solid tumors (as noted in the Introduction), it is not clear why a TET1 inhibitor would be expected to show decreasing viability proportional to TET1 expression across all 60 lines. This is important, again, inasmuch as the question of non-specific toxicity needs to be considered carefully. It may be worthwhile to show the primary data from this analysis.
2. The authors make the case that NSC-311068 and NSC-370284 decrease the expression of TET1 mRNA. However, at the time point (48 hours) and concentration (300 nM) that they examine (Figure 1c), there is >90% loss of viability of sensitive cells, making the significance of this finding difficult to determine. This likely also applies to Figure 2c and d. Many mRNAs and proteins with short half lives will decrease after cells are exposed to toxic agents, but that does not mean that these decreases are specific or driving the change in viability.
3. The suggestion that STAT transcription factors may be involved in the mechanism of action of

these compounds derived from a mutational analysis related to clones resistant to these compounds. The authors then extrapolated from pathway analysis that the "Jak/STAT5" pathway may be involved. However, it is not apparent that any of the mutations shown in Supplemental Table 3 would overcome the effect of a direct STAT inhibitor (the proposed mechanism).

4. Figure 3 (and subsequent figures) shows data on STAT DNA binding. It would be worthwhile to show the levels of STAT3 and STAT5 phosphorylation in the relevant cells, and how they change, if at all (particularly at early time points), with these compounds.

5. If the mechanism of action of these compounds is the inhibition of STAT3 and/or STAT5 transcriptional function, it is not at all clear why the other STAT inhibitors they tested failed to show a similar effect. Conversely, since many other genes necessary for proliferation and survival are STAT-dependent, it is not clear why these compounds would only be effective in TET1-high AML cells. The authors should also note if there is any evidence that these compounds inhibit signatures of STAT-dependent genes in these AML cells, as would be expected if they exert a STAT-dependent mechanism.

6. The authors show that transduction of MLL_AF9 into bone marrow progenitor cells confers sensitivity to the compounds of interest (Figure 5). Since this transduced gene is presumably no longer under the control of a STAT-dependent promoter, this would seem to undermine the central hypothesis regarding the mechanism of action of the compound.

7. A plausible mechanism for why the resistant subclones are sensitive to daunorubicin should be proposed. It is not at all obvious why this should be so.

8. If the underlying mechanism for these compounds effects is that they downregulate TET1, then ectopic expression of TET1 should rescue cells from their effects. This experiment would seem worthwhile to perform.

SUMMARY: We are very grateful to the reviewers for their positive appraisals of our manuscript and very much appreciate their thoughtful and constructive suggestions and comments. Following their suggestions and comments, we have performed additional experiments and data analysis, and have also carefully revised the manuscript accordingly. Summarized below are major experiments and data analysis conducted for the paper revision:

- a) A series of additional experiments have been carried out to further strengthen the mechanism studies of NSC-370284. We first created an inducible *Tet1* construct, and found ectopic expression of *Tet1* sufficiently reversed the inhibitory effects of NSC-370284 and UC-514321 and restored viability of the AML cells. To understand why NSC-370284 exhibited a selective effect on targeting *TET1*-high AMLs, we analyzed the affinity of STAT3/5 on the promoter regions of *TET1* and other STAT target genes, e.g., *HIF2 α* , by ChIP-qPCR, and found that the STAT proteins have a stronger enrichment on the promoter of *TET1* than on that of *HIF2 α* in *TET1*-high AML cells and NSC-370284 could dramatically inhibit the binding of STAT proteins to *TET1* promoter. Further, we detected no significant influences of NSC-370284 on STAT activity (i.e., phosphorylation). Our results indicate that different from typical STAT inhibitors that target STAT kinase activity, NSC-370284 (and its derivative, UC-514321) may exert its function mainly through interfering with the association between STAT protein and the DNA regions which have relatively higher basal affinity to STATs, like the *TET1* promoter, in AML cells.
- b) We were able to explore into the mechanisms underlying NSC-370284 resistance in more details as well. Choosing *JAK1*^{A893G}, one of the mutations associated with NSC-370284 resistance as a representative, we showed ectopic expression of this mutant largely reversed the inhibitory effect of NSC-370284 on *MLL-AF9*-AML cell viability. It is likely the mutations in genes associated with the JAK/STAT signaling that were identified in our drug-resistant clones may overcome NSC-370284-mediated inhibitory effect on activation of the JAK/STAT signaling and AML cell viability/growth, and thereby confer drug resistance to AML clones.
- c) Two more AML models, i.e., *MLL-AF10* AML and *FLT3-ITD/NPM1*^{mut} AML, have been included to show the broad therapeutic effects of our lead compounds in treating AML.
- d) Additional cell viability assays have been conducted to assess the effects of NSC-370284 and UC-514321 in normal hematopoietic stem/progenitor cells (HSPCs; c-Kit⁺ BM cells) *in vitro*. Our data showed that while the compounds significantly inhibited the viability and increased apoptosis of *Tet1*-high AML cells, they showed minor effects on normal HSPCs, suggesting a good therapeutic window for the application of our compounds in treating *TET1*-high AMLs.
- e) More preclinical data, including maximum tolerated dose (MTD), LD50 and *in vivo* preclinical pharmacokinetic (PK) analysis, have been added.
- f) More analysis of the RNA-seq data of NSC-370284-treated or NSC-370284-resistant AML cells have been included, to show the genes and enriched gene clusters associated with daunorubicin (DNR) sensitivity.

All the text in the manuscript that we changed or added during the revision is marked in red. Deleted text has been removed entirely from the file. Our detailed responses are described below. We hope that the revised version of the manuscript would fulfill the requirements of the reviewers and editor, and will be suitable for the publication in *Nature Communications*.

The following are our point-by-point responses to the comments/suggestions from the reviewers:

1. Responses to the comments of reviewer 1:

General Comments from Reviewer 1: *TET1 is the founding member of a family of DNA methylcytosine dioxygenases (including TET1/2/3), which are known to convert 5-methylcytosine (5mC) to 5-hydroxymethylcytosine (5hmC), leading to DNA demethylation. While previous studies suggested that all three TET genes likely function as tumor-suppressor genes in various cancers, the Chen lab and others have provided compelling evidence showing that TET1 plays a critical oncogenic role in the development of myeloid malignancies. Importantly, Tet1 knockout has been reported to have minor effects on normal development including hematopoiesis in mice. Therefore, TET1 is a promising therapeutic target for AML treatment and targeting TET1 signaling is a hot pursuit in the field of cancer/myeloid malignancies. In this manuscript, Jiang et al. timely reported the identification of several novel therapeutic drugs against TET1 signaling. The authors demonstrated the high efficacy and selectivity of these drugs in targeting acute myeloid leukemias (AMLs) with high levels of TET1 expression (i.e., TET1-high AMLs) both in vitro and in vivo. To develop inhibitors targeting TET1 signaling, the authors employed an innovative strategy to screen small-molecule compounds that inhibit the transcription of TET1, rather than inhibiting TET1's enzymatic activity, since TET1 exerts its oncogenic role through both enzymatic-dependent and -independent mechanisms. Using this strategy, the authors identified compounds NSC-311068 and NSC-370284 as selective inhibitors of TET1 signaling by inhibiting TET1 transcription. They showed that both compounds, especially NSC-370284, exhibited potent therapeutic effects in treating TET1-high AMLs. A structural analogue of NSC-370284, namely UC-514321, exhibited even better therapy effects. In addition, the authors delineated the molecular mechanism by which these compounds inhibit the TET1 transcription, by showing that the compounds directly target the transcriptional activators of TET1, STAT3/5. The authors also showed that NSC-370284 or UC-514321 exhibit a synergistic therapeutic effect with standard chemotherapy agents such as daunorubicin. While, these compounds had no noticeable (both acute and long-term) side effects on normal hematopoietic system in vivo, highlighting the low toxicity of these compounds for future potential clinical application. In summary, this is the first report for the identification of therapeutic agents targeting TET1 signaling, which exhibit high efficacy and selectivity for treating TET1-high AMLs. Moreover, it is novel to show that STAT3/5 are direct transcriptional regulators of TET1 and small-molecule inhibitors targeting STAT3/5 can efficiently suppress TET1 expression. This work is of great significance and novelty in both basic and clinical cancer research. The combination of such TET1 inhibitors with standard chemotherapy agents holds potential to improve the clinical treatment of TET1-high AMLs.*

Response: We appreciate the reviewer's highly positive appraisal of our manuscript.

Comment 1 (Minor points) from Reviewer 1: *While the therapeutic effects of the TET1 transcription inhibiting compounds are quite promising, the authors should provide more pre-clinical data such as PK/PD, which are important to evaluate the potential of clinical appliances.*

Response: We appreciate reviewer's suggestion. With the help of pharmacological collaborators, we have now included maximum tolerated dose (MTD), LD50 and *in vivo* preclinical pharmacokinetic (PK) analysis. The maximum tolerated dose of NSC-370284 and UC-514321 in mice was 65~85.6 mg/kg (**Supplementary Table 8**). The LD50 was around 123 mg/kg

(**Supplementary Table 8**). Analysis of the pharmacokinetic properties of UC-514321 showed that the compound had a half-life of 11.02 hrs in mouse blood (**Supplementary Table 9; Supplementary Fig. 11**).

Comment 2 (Minor points) from Reviewer 1: Although the authors have shown that these TET1 transcription inhibiting compounds exhibited no obvious effects on normal hematopoiesis in vivo, it would be helpful to examine the effects of these compounds on normal HSC/HPCs in vitro, if possible compared with TET1-high HSC/HPCs. Colony-forming assay and cell viability (apoptosis) assay can be applied.

Response: We thank reviewer for the very good suggestion. Cell viability and apoptosis assays have been carried out to assess the effects of NSC-370284 and UC-514321 on normal hematopoietic stem/progenitor cells (HSPCs; herein we used c-Kit⁺ BM cells) *in vitro*. Remarkably, NSC-370284 or UC-514321 treatment dramatically suppressed the viability of AML cells, but not that of normal HSPCs (**Supplementary Fig. 8a**). In addition, the compounds significantly increased apoptosis in AML cells, but not in normal HSPCs (**Supplementary Fig. 8b**). Thus, these two compounds showed no obvious toxicity on normal HSPCs. This is consistent with endogenous *Tet1* expression pattern, as AML cells with *MLL-AF10* or *FLT3-ITD/NPM1^{mut}* have relatively higher *Tet1* expression levels, as compared with normal HSPCs (**Supplementary Fig. 8c**). The above results have been included as the new **Supplementary Fig. 8**.

2. Responses to the comments of reviewer 2:

General Comments from Reviewer 2: Alteration in enzymatic DNA methylcytosine oxidation has been reported in AML. But the therapeutic opportunity has not been explored. This manuscript describes the first effort in suppressing AML by inhibiting the function of TET oxidase which appears to act as an oncogene. The overall concept presented in this manuscript is novel. More and more basic transcriptional machinery factors are now being explored for therapeutic applications with success and promises, such as BRD4 inhibitors (best example), integrator complex, TAF250 and so on. The inhibitors reported in this work actually targeted STATs but was identified by using TET1 as readout in a screen. If these compounds had reliable efficacy to AML cells in vivo and not much side effects, they would be promising in principle. However, the specificity to TET1 transcription inhibition has to be further studied. The question whether the compounds deliver their effects really through inhibiting TET1 transcription is not fully addressed. Targeting Stats would influence transcription of many genes. Additionally, AML is highly heterogenous in its genetics, therefore the mechanism that makes the compounds specifically effective for the AML with high TET1 expression would need to be better characterized.

Response: We appreciate the reviewer's overall positive appraisal of our manuscript and we are very grateful to him/her for the constructive comments and suggestions. We have taken the reviewer's advice and performed additional experiments to further address potent therapeutic effects of the candidate compounds and to better characterize the molecular mechanism underlying their selectivity on targeting *TET1*-high AML cells. These new results have been added into this revised manuscript.

Comment 1 from Reviewer 2: *A specific suggestion is to do inducible ectopic expression of TET1 in AML cells treated with UC-514321 and examine whether this would mitigate the effect of the compound, restoring cell viability.*

Response: Following the reviewer's suggestion, we created a pLenti-puro vectored *Tet1* construct, and transduced it into THP-1 and MONOMAC-6 cells and showed that ectopic expression of *Tet1* sufficiently reversed the inhibitory effects of NSC-370284 and UC-514321 and restored viability of the AML cells. The new results have been included as **Figs. 5n-o**.

Comment 2 from Reviewer 2: *The compounds appeared to work well in vivo with the MLL-AF9 and AML-ETO9a models. But how is their effect with other important models?*

Response: Reviewer raised an important question. We expanded the test of the therapeutic effects of NSC-370284 and UC-514321 into two additional AML models, i.e., *MLL-AF10* AML and *FLT3-ITD/NPM1^{mut}* AML. As expected, NSC-370284 prolonged median survival of *MLL-AF10* leukemic mice from 35 days to >50 days, and that of *FLT3-ITD/NPM1^{mut}* leukemic mice from 21 days to 34 days. UC-514321 showed an even better therapeutic effect, as it prolonged median survival of both *MLL-AF10* and *FLT3-ITD/NPM1^{mut}* leukemic mice to >50 days. Results have been included as **Figs. 4i and 4j**.

Comment 3 from Reviewer 2: *The title appears somewhat confusing as the compounds bind STAT3/5 as STAT inhibitors rather than inhibit TET1 transcription directly.*

Response: We appreciate reviewer's comment, and have changed the title to "**Targeted inhibition of STAT/TET1 axis as a potent therapeutic strategy for acute myeloid leukemia**".

3. Responses to the comments of reviewer 3:

General Comments from Reviewer 3: *In this manuscript, the authors focus on TET1 as a target in AML. Based on a chemical library screen database, they have identified several compounds that they suggest decrease TET1 expression in a STAT3- or STAT5-dependent manner, and exert a therapeutic effect in in vitro and murine models of AML.*

The overall concept of this study is very interesting and provocative, and these compounds certainly do show activity in the experiments shown. However, the data leave open many fundamental questions regarding the mechanism by which these compounds are exerting their effect, and much of the data can be explained by non-specific toxicity, especially as at least one of the compounds is structurally related to a known topoisomerase II inhibitor.

Response: We appreciate the reviewer's positive appraisal of our manuscript. Following the reviewer's constructive suggestions and comments, we have conducted additional experiments to further strengthen the functional evaluation and mechanistic studies of the chemical compounds. New results have been included in **Figs. 1, 4, 5, Supplementary Figs.1, 4, 5, 9, 10, 11, and Supplementary Tables 2, 3, 8, 9 and 10**.

Comment 1 from Reviewer 3: *The rationale for the strategy to identify the initial lead compounds should be clarified. In the NCI60 panel, only six of the lines are hematopoietic. Since TET1 may have a tumor suppressor role in solid tumors (as noted in the Introduction), it is not*

clear why a TET1 inhibitor would be expected to show decreasing viability proportional to TET1 expression across all 60 lines. This is important, again, in as much as the question of non-specific toxicity needs to be considered carefully. It may be worthwhile to show the primary data from this analysis.

Response: We thank the reviewer for the comment. We previously published a series of research articles and reported the high expression and oncogenic role of *TET1* in AML (Huang et al., *Proc Natl Acad Sci USA*, 2013; Huang et al., *Cancer Lett*, 2016; Jiang et al., *Nat Commun*, 2016). In fact, high expression of *TET1* could be found not only in AML, but also in many other tumors, e.g., uterine cancer, glioma, etc., and especially, in testicular germ cell malignancies. This indicates potential oncogenic role of *TET1* in various cancers. We have included the TCGA expression pattern of *TET1* as new **Supplementary Fig. 1**. The original drug response data has been enclosed as the new **Supplementary Table 2**.

Comment 2 from Reviewer 3: The authors make the case that NSC-311068 and NSC-370284 decrease the expression of TET1 mRNA. However, at the time point (48 hours) and concentration (300 nM) that they examine (Figure 1c), there is >90% loss of viability of sensitive cells, making the significance of this finding difficult to determine. This likely also applies to Figure 2c and d. Many mRNAs and proteins with short half lives will decrease after cells are exposed to toxic agents, but that does not mean that these decreases are specific or driving the change in viability.

Response: This is a good point. In order to rule out the possibility of non-specific toxicity, we reduced the dose of NSC-311068 and NSC-370284 to 25 nM, and analyzed gene expression and cell viability 24 hours after treatment. The low dose, short-term treatments again resulted in a significant down-regulation of *TET1* transcription, accompanied with a very minor decrease in the viability of MONOMAC-6, THP-1 and KOCL-48 cells (see new **Fig. 1e-f**). Thus, it is unlikely that the inhibitory effects of NSC-311068 and NSC-370284 on *TET1* expression were due to nonspecific toxicity. Shown in **Figs. 2c-d** are results of *in vivo* sample analysis. Briefly, we conducted secondary BMT, treated the leukemic mice with the chemical compounds or control, collected BM samples at their end points and performed statistical analysis of survival data and conducted qPCR and Western blot assays to evaluate the effects of drug treatment on *Tet1* expression. The dose of NSC-311068 and -370284 treatment (i.e., 2.5 mg/kg, i.p., once per day, for 10 days) has been proven to be within the safe range, based on the acute and long-term toxicity profiling, the maximum tolerated dose, LD50 and *in vivo* preclinical pharmacokinetic (PK) analysis shown in **Supplementary Tables 7-9**, and **Supplementary Figs. 9-11**. Thus, it is unlikely the inhibitory effects of NSC-311068 and NSC-370284 on *TET1* expression were due to nonspecific toxicity.

Comment 3 from Reviewer 3: The suggestion that STAT transcription factors may be involved in the mechanism of action of these compounds derived from a mutational analysis related to clones resistant to these compounds. The authors than extrapolated from pathway analysis that the “Jak/STAT5” pathway may be involved. However, it is not apparent that any of the mutations shown in Supplemental Table 3 would overcome the effect of a direct STAT inhibitor (the proposed mechanism).

Response: We thank the reviewer for the comment. Indeed, the mechanism underlying drug resistance is often complicated and remains a challenge in the field of cancer research. A number

of the genes identified from our drug-resistant AML clones with recurrent mutations, e.g., *MSH3*, *Notch*, etc., have been reported to be associated with the JAK/STAT signaling (Tseng-Rogenski et al., *Gastroenterology*, 2015; Cheng et al., *Cellular Signalling*, 2010; Gagarin et al., *J Mol Cell Cardiol*, 2005; Huang et al., *Cell Cycle*, 2008). It is likely that mutations in such genes may overcome NSC-370284-mediated inhibitory effect on activation of the JAK/STAT signaling and AML cell viability/growth, and thereby confer drug resistance to the AML clones. To test this, we chose *JAK1* as a representative and cloned a construct carrying the *JAK1*^{A893G} mutant that was detected in our drug-resistant THP-1 cells (see **Supplementary Table 4**). As expected, forced expression of *JAK1*^{A893G} mutant largely reversed the inhibitory effect of NSC-370284 on AML cell viability (**Supplementary Fig. 4c**). It is out of the scope of this paper for us to test pathological effects of all the mutations, but investigating these drug resistance-associated mutations would definitely be an interesting research topic in the future.

Comment 4 from Reviewer 3: Figure 3 (and subsequent figures) shows data on STAT DNA binding. It would be worthwhile to show the levels of STAT3 and STAT5 phosphorylation in the relevant cells, and how they change, if at all (particularly at early time points), with these compounds.

Response: Following the reviewer's suggestion, we treated MONOMAC-6 cells with 250 nM NSC-370284 for 0 min, 15 min, 30 min, 2 hrs, and 24 hrs. Western blotting showed no significant alterations of the phosphorylation levels of STAT3 and STAT5. Our data suggest that it is likely NSC-370284 mainly competes against DNA for STAT binding, but not suppresses STAT phosphorylation. The above results have been included as Supplementary Fig. 4h.

Comment 5 from Reviewer 3: If the mechanism of action of these compounds is the inhibition of STAT3 and/or STAT5 transcriptional function, it is not at all clear why the other STAT inhibitors they tested failed to show a similar effect. Conversely, since many other genes necessary for proliferation and survival are STAT-dependent, it is not clear why these compounds would only be effective in TET1-high AML cells. The authors should also note if there is any evidence that these compounds inhibit signatures of STAT-dependent genes in these AML cells, as would be expected if they exert a STAT-dependent mechanism.

Response: We appreciate the reviewer's comment. We have now also analyzed the inhibitory effect of NSC-370284 on the expression of *HIF2α*, a well known target of STAT5, and found that the inhibitory effect was not as obvious as that on *TET1* (**Supplementary Fig. 4i**). The basal association of STAT5 with the *HIF2α* promoter was low, as compared to that with the *TET1* promoter region; and the interruption by NSC-370284 on such association was much less obvious (**Supplementary Fig. 4j**). The very weak, if any, basal affinity of STAT5 with many of its target genes' promoters (e.g., *BCL-x*, *HIF2α*) without cytokine stimulation (e.g., IL-3 or EPO) has been reported before (Nelson et al., *J Biol Chem*, 2004; Fatrai et al., *Blood*, 2011). Therefore, our results revealed a particularly strong association between STAT3/5 and the *TET1* promoter (**Fig. 3e-g, 3k; Supplementary Fig. 4i-j**). Our results indicate that different from typical STAT inhibitors that target STAT kinase activity, NSC-370284 (and its derivative, UC-514321) may exert its function mainly through interfering with the association between STAT protein and the DNA regions which have relatively higher basal affinity to STATs, like the *TET1* promoter, in AML cells. The new data and above discussion have been added into the revised manuscript.

Comment 6 from Reviewer 3: The authors show that transduction of MLL_{AF9} into bone marrow progenitor cells confers sensitivity to the compounds of interest (Figure 5). Since this transduced gene is presumably no longer under the control of a STAT-dependent promoter, this would seem to undermine the central hypothesis regarding the mechanism of action of the compound.

Response: Yes, ectopic expression of *MLL-^{AF9}* is not STAT-dependent. However, *MLL-^{AF9}*-mediated upregulation of endogenous *Tet1* expression, as reported previously by us (Huang H, et al. *PNAS*, 2013) and also shown in **Supplementary Fig. 8c**, is highly likely still STAT-dependent and thus can be significantly inhibited by our compounds, which in turn leads to the significant inhibitor effect of the compounds on the viability of *MLL-^{AF9}*-transduced bone marrow progenitor cells as shown in new **Figs. 5i-m**. Therefore, such data actually support our central hypothesis regarding the mechanism of action of our compound.

Comment 7 from Reviewer 3: A plausible mechanism for why the resistant subclones are sensitive to daunorubicin should be proposed. It is not at all obvious why this should be so.

Response: We appreciate the reviewer's suggestion. Through analysis of the RNA-seq data of the NSC-370284 resistant clones and parental cells, we found that several gene clusters that are known to be associated with drug response, especially response to topoisomerase II inhibitors such as DNR, are enriched in NSC-370284 resistant cells. These gene clusters include JAK/STAT signaling, G2M checkpoint, MYC targets and E2F targets, etc. (**Supplementary Table 10a**). It was reported that JAK/STAT pathway inhibitors, e.g. AG490, could sensitize tumor cells to topoisomerase II inhibitors¹. A potential mechanism might be through targeting E2F. It was shown that *E2F1* overexpression significantly sensitized cancer cells to various topoisomerase II inhibitors². Our RNA-seq data showed increased *E2F1* levels in NSC-370284 resistant AML clones, relative to parental cells. This is consistent with the E2F1-inducing effect of other STAT inhibitors, e.g. AG490³. Also consistent is the enrichment of E2F targets in control samples relative to NSC-370284 treated samples (**Supplementary Table 10b**). Therefore, very likely the upregulation of *E2F1* in NSC-370284 resistant clones at least partially explains why the resistant clones, compared to the parental AML cells, are more sensitive to DNR treatment. The new data and above discussion have been included in the revised manuscript now.

Comment 8 from Reviewer 3: If the underlying mechanism for these compounds effects is that they downregulate TET1, then ectopic expression of TET1 should rescue cells from their effects. This experiment would seem worthwhile to perform.

Response: As described in our response to Comment 1 from Reviewer 2, we have created a pLenti-puro vectored *Tet1* construct, and transduced it into THP-1 and MONOMAC-6 cells and showed that ectopic expression of *Tet1* sufficiently reversed the inhibitory effects of NSC-370284 and UC-514321 and restored viability of the AML cells (**Fig. 5n-o**). This new data together with other relevant data from our paper provide compelling evidence supporting our proposed mechanism underlying the selective effects of the compounds on *TET1*-high AML cells.

References cited:

1. Gariboldi MB, Ravizza R, Molteni R, Osella D, Gabano E, Monti E. Inhibition of Stat3 increases doxorubicin sensitivity in a human metastatic breast cancer cell line. *Cancer Lett* **258**, 181-188 (2007).
2. Dong YB, Yang HL, Elliott MJ, McMasters KM. Adenovirus-mediated E2F-1 gene transfer sensitizes melanoma cells to apoptosis induced by topoisomerase II inhibitors. *Cancer Res* **62**, 1776-1783 (2002).
3. Savell J, *et al.* AG490 inhibits G1-S traverse in BALB/c-3T3 cells following either mitogenic stimulation or exogenous expression of E2F-1. *Mol Cancer Ther* **3**, 205-213 (2004).

Reviewers' comments:

Reviewer #1 (Remarks to the Author):

The authors adequately addressed all my concerns and the revised manuscript is dramatically improved. The conclusions are better supported by the data now. I have no further comments for this revised manuscript.

Reviewer #2 (Remarks to the Author):

My concerns have been well addressed.

Reviewer #3 (Remarks to the Author):

This revised version leaves many significant unanswered questions, that were not adequately addressed in the response. To name just four:

1. Given that TET1 acts as a tumor suppressor in many solid tumors, there is no satisfactory explanation for why these compounds, should they be acting in a TET1-specific manner, would appear active in an NCI60 viability screen (comment 1).
2. There is no satisfactory explanation of why mutations upstream in the STAT signaling pathway would overcome the effect of a compound that blocks STAT binding to DNA (comment 3).
3. A convoluted response is provided as to why their STAT inhibitors show this therapeutic effect, but other STAT inhibitors do not. This is based on the hypothesis that the binding of STAT5 to the TET1 binding site may be higher than that of other sites (although they only seem to look at one other site). This still would not explain why a compound that completely blocks STAT activity would not have a similar effect (comment 5).
4. To explain why cells resistant to their inhibitor become sensitive to daunorubicin, they make the argument that other STAT3 inhibitors have this effect. Thus, the question is whether STAT3 is still being inhibited in the resistant cells (a very different question). This key point is never addressed (comment 7).

We are very grateful to the reviewers for the positive appraisals from Reviewers 1 and 2, and very much appreciate constructive suggestions and comments from Reviewer 3. Following these suggestions and comments, we have performed additional experiments and data analysis, and have also carefully revised the manuscript accordingly. Listed below are our point-by-point responses to the comments/suggestions from the reviewers:

1. Response to Reviewer 1:

General Comment: *The authors adequately addressed all my concerns and the revised manuscript is dramatically improved. The conclusions are better supported by the data now. I have no further comments for this revised manuscript.*

Response: We thank Reviewer 1 for the positive appraisals.

2. Response to Reviewer 2:

General Comment: *My concerns have been well addressed.*

Response: We thank Reviewer 2 for the affirmation.

3. Response to Reviewer 3:

General Comment: *This revised version leaves many significant unanswered questions, that were not adequately addressed in the response.*

Response: We thank the Reviewer for the further suggestions and comments! As described in the following point-by-point response, we have conducted additional experiments and data analysis, and have also carefully revised the manuscript accordingly.

Specific Comment 1: *1. Given that TET1 acts as a tumor suppressor in many solid tumors, there is no satisfactory explanation for why these compounds, should they be acting in a TET1-specific manner, would appear active in an NCI60 viability screen (comment 1).*

Response: We thank the Reviewer for the comment. Actually, from the beginning we already expected that the inhibitory effects (on tumor-cell growth/viability) of such compounds and *TET1* expression levels would not be positively correlated in all individual cells lines of the NCI-60 panel due to noises caused by the potential tumor-suppressor effects of *TET1* in certain types of solid tumors. Indeed, as reported in the literature¹⁻⁵, *TET1* functions as a tumor suppressor in certain types of solid tumors in which *TET1* expression is at very low levels or silenced. Nevertheless, our goal was to identify the particular compounds that exhibit the most robust positive correlation between their inhibitory activities and *TET1* expression levels in *TET1*-high cancer cell lines, and such positive correlation should be able to override or surpass the expected noise from the cell lines where *TET1* is expressed at low levels and thereby endow a statistically positive correlation to the entire NCI-60 panel.

Indeed, as expected, when we re-analyzed the NCI-60 dataset, we found that cancer cell lines with relatively higher *TET1* levels showed a more obvious positive correlation between *TET1* expression levels and activities of both NSC-311068 and NSC-370284, compared to that across the entire NCI-60 panel, whereas cell lines with relatively lower *TET1* level exhibited no obvious positive correlation (NSC-311068) or an even negative correlation (NSC-370284). Therefore, our data further suggest that the anti-tumor effects of NSC-311068 and NSC-370284 heavily rely on *TET1* expression levels. Our new analysis results have now been included as **Supplementary Table 2c-d** and we have also described such results in the main text.

Specific Comment 2: *2. There is no satisfactory explanation of why mutations upstream in the STAT signaling pathway would overcome the effect of a compound that blocks STAT binding to DNA (comment 3).*

Response: Reviewer raised a good point. As described below, we have now performed three major experiments to address this concern.

1) We constructed one more construct carrying another point mutation, namely *MSH3*^{V600I}, and tested its role in NSC-370284 response. Similar to *JAK1*^{A893G}, forced expression of the *MSH3*^{V600I} mutant also partially reversed the inhibitory effect of NSC-370284 on AML cell viability. Results of *MSH3*^{V600I} have been included as new **Supplementary Fig. 4d**.

2) Interestingly, we found that knockdown of *TET1* resulted in a reduction of *JAK1* transcription in AML cells (**Supplementary Fig. 4i**). This indicates potential regulation of TET1 on *JAK1* expression.

3) ChIP-qPCR results show direct binding of TET1 to the *JAK1* promoter (**Supplementary Fig. 4j**). The above findings suggest JAK/STAT pathway promotes *TET1* transcription via direct binding of STAT3/5 to the *TET1* promoter, and TET1 also binds to the *JAK1* promoter and activates *JAK1* transcription. Herein we unveil a feedback loop between JAK1/STAT/TET1 in AML cells.

Therefore, our data suggest that mutations in JAK1, MSH3, and/or some other genes belonging to the JAK/STAT network (see **Fig. 3a**) likely contribute to the drug resistance of the individual (drug-resistant) AML clones.

Specific Comment 3: *3. A convoluted response is provided as to why their STAT inhibitors show this therapeutic effect, but other STAT inhibitors do not. This is based on the hypothesis that the binding of STAT5 to the TET1 binding site may be higher than that of other sites (although they only seem to look at one other site). This still would not explain why a compound that completely blocks STAT activity would not have a similar effect (comment 5).*

Response: As we discussed in the manuscript, while other compounds were designed to block the phosphorylation activity of STAT proteins, the compounds we identified exhibit an unique character, i.e., selectively inhibiting the binding of STAT protein to *TET1* promoter region and thereby exhibiting a more selective and robust inhibition on *TET1* transcription in *TET1*-high AML cells. In contrast, other compounds show an anti-tumor effect less dependent on TET1 levels.

To further validate the stronger binding of STAT proteins to the *TET1* promoter than to the promoters of other target genes of STAT proteins, we tested two more known target genes of STAT5, i.e., *IL2RA*⁶ and *FRA2*⁷, besides the previously tested *HIF2 α* . The inhibitory effect of NSC-370284 on the expression of all these three target genes was not as potent as that on *TET1* expression (**Supplementary Fig. 4l**). The basal enrichment of STAT5 on the promoter of *HIF2 α* , *IL2RA* or *FRA2* was much lower than that on the *TET1* promoter; and the interruption by NSC-370284 on such association was much less obvious (**Supplementary Fig. 4m**). The very weak, if any, basal affinity of STAT5 with most of its target genes' promoters without cytokine stimulation (e.g., IL-2, IL-3 or EPO) has been reported before⁶⁻⁹.

Therefore, our results indicate that our compounds exhibit a more selective and robust inhibition on *TET1* transcription through inhibiting the binding of STAT3/5 to *TET1* promoter region (**Fig. 3e-g, 3k; Supplementary Fig. 4l-m**).

Specific Comment 4: *4. To explain why cells resistant to their inhibitor become sensitive to daunorubicin, they make the argument that other STAT3 inhibitors have this effect. Thus, the question is whether STAT3 is still being inhibited in the resistant cells (a very different question). This key point is never addressed (comment 7).*

Response: We appreciate reviewer's comment, and have revised this part of analysis and discussion accordingly. Indeed, STAT pathways were enriched in NSC-370284-resistant clones, relative to parental AML cells (**Supplementary Table 10a**), suggesting that the STAT3/5 signaling is not inhibited in the resistant cells.

Now, we proposed a new model to explain why cells resistant to our inhibitor become sensitive to daunorubicin. We think that this might be through targeting G2M checkpoint, one of the gene sets enriched in NSC-370284 resistant AML clones (**Supplementary Table 10a**). It was reported previously that overexpression of *CDC25*, a key phosphatase of G2M checkpoint control, could significantly sensitize tumor cells to doxorubicin treatment¹⁰. Our RNA-seq data showed increased *CDC25* levels in NSC-370284 resistant AML clones, relative to the parental cells. Also consistent is the enrichment of G2M checkpoint gene cluster in control samples relative to NSC-370284 treated samples (**Supplementary Table 10b**). The activation of JAK/STAT pathway might, directly or indirectly, contribute to G2M checkpoint abnormality, as reported previously by others^{11,12}. Therefore, very likely the deregulation of G2M checkpoint in NSC-370284 resistant clones at least partially explains why the resistant clones, compared to the parental AML cells, are more sensitive to DNR treatment.

References Cited:

1. Haffner, M.C., *et al.* Global 5-hydroxymethylcytosine content is significantly reduced in tissue stem/progenitor cell compartments and in human cancers. *Oncotarget* **2**, 627-637 (2011).
2. Yang, H., *et al.* Tumor development is associated with decrease of TET gene expression and 5-methylcytosine hydroxylation. *Oncogene* **32**, 663-669 (2013).
3. Lian, C.G., *et al.* Loss of 5-hydroxymethylcytosine is an epigenetic hallmark of melanoma. *Cell* **150**, 1135-1146 (2012).

4. Hsu, C.H., *et al.* TET1 Suppresses Cancer Invasion by Activating the Tissue Inhibitors of Metalloproteinases. *Cell Rep* **2**, 568-579 (2012).
5. Sun, M., *et al.* HMGA2/TET1/HOXA9 signaling pathway regulates breast cancer growth and metastasis. *Proc Natl Acad Sci USA* **110**, 9920-9925 (2013).
6. Nagy, Z.S., *et al.* Genome wide mapping reveals PDE4B as an IL-2 induced STAT5 target gene in activated human PBMCs and lymphoid cancer cells. *PLoS One* **8**, e57326 (2013).
7. Rani, A., Greenlaw, R., Runglall, M., Jurcevic, S. & John, S. FRA2 is a STAT5 target gene regulated by IL-2 in human CD4 T cells. *PLoS One* **9**, e90370 (2014).
8. Nelson, E.A., Walker, S.R., Alvarez, J.V. & Frank, D.A. Isolation of unique STAT5 targets by chromatin immunoprecipitation-based gene identification. *J Biol Chem* **279**, 54724-54730 (2004).
9. Fatrai, S., Wierenga, A.T., Daenen, S.M., Vellenga, E. & Schuringa, J.J. Identification of HIF2alpha as an important STAT5 target gene in human hematopoietic stem cells. *Blood* **117**, 3320-3330 (2011).
10. Varmeh, S. & Manfredi, J.J. Overexpression of the dual specificity phosphatase, Cdc25C, confers sensitivity on tumor cells to doxorubicin-induced cell death. *Mol Cancer Ther* **7**, 3789-3799 (2008).
11. Barry, S.P., *et al.* STAT3 modulates the DNA damage response pathway. *Int J Exp Pathol* **91**, 506-514 (2010).
12. Zheng, Y., *et al.* A CK2-dependent mechanism for activation of the JAK-STAT signaling pathway. *Blood* **118**, 156-166 (2011).